# Dynamic trafficking and turnover of JAM-C is essential for endothelial cell migration

**Katja B. Kostelnik**[1©], **Amy Barker**[1©], **Christopher Schultz**[1], **Tom P. Mitchell**[1], **Vinothini Rajeeve**[2], **Ian J. White**[3], **Michel Aurrand-Lions**[4], **Sussan Nourshargh**[1], **Pedro Cutillas**[2], **Thomas D. Nightingale**[1]*

**1** Centre for Microvascular Research, William Harvey Research Institute, Barts and the London School of Medicine and Dentistry, Queen Mary University of London, United Kingdom, **2** Cell Signalling & Proteomics Group, Barts Cancer Institute, Queen Mary University of London, London, United Kingdom, **3** MRC Laboratory of Molecular Cell Biology, University College London, London, United Kingdom, **4** Aix Marseille University, CNRS, INSERM, Institut Paoli-Calmettes, CRCM, Marseille, France

© These authors contributed equally to this work.
* t.nightingale@qmul.ac.uk

**Data Availability Statement:** All relevant data are included in the paper and its supporting files. In addition, the mass spectrometry proteomics data have been deposited to the ProteomeXchange

## Abstract

Junctional complexes between endothelial cells form a dynamic barrier that hinders passive diffusion of blood constituents into interstitial tissues. Remodelling of junctions is an essential process during leukocyte trafficking, vascular permeability, and angiogenesis. However, for many junctional proteins, the mechanisms of junctional remodelling have yet to be determined. Here, we used receptor mutagenesis, horseradish peroxidase (HRP), and ascorbate peroxidase 2 (APEX-2) proximity labelling, alongside light and electron microscopy (EM), to map the intracellular trafficking routes of junctional adhesion molecule-C (JAM-C). We found that JAM-C cotraffics with receptors associated with changes in permeability such as vascular endothelial cadherin (VE-Cadherin) and neuropilin (NRP)-1 and 2, but not with junctional proteins associated with the transmigration of leukocytes. Dynamic JAM-C trafficking and degradation are necessary for junctional remodelling during cell migration and angiogenesis. By identifying new potential trafficking machinery, we show that a key point of regulation is the ubiquitylation of JAM-C by the E3 ligase Casitas B-lineage lymphoma (CBL), which controls the rate of trafficking versus lysosomal degradation.

## Introduction

Endothelial cells exist as a monolayer that lines the blood vasculature, and as such, they present a discrete barrier to the cellular and molecular components of blood. This includes ions, small and large proteins, platelets, and leukocytes [1]. The endothelium also plays a crucial role in responding to a pathogenic infection or tissue injury, both in the initial recruitment of leukocytes to the appropriate site [2] and in subsequent tissue repair and angiogenesis [3]. Endothelial dysfunction can predispose the vessel wall to leukocyte adhesion, platelet activation, oxidative stress, thrombosis, coagulation, and chronic inflammation, leading to the pathogenesis of numerous cardiovascular diseases [4].

Consortium via the PRIDE partner repository with the dataset identifier PXD013003.

**Funding:** TDN, CS, and KBK were funded by an MRC project grant MR/M019179/1. KBK also received funding from the People Programme (Marie Curie Actions) of the European Union's Seventh Framework Programme (FP7/2007-2013) under REA grant agreement n° 608765. AB and TPM were funded by QMUL. SN was funded by a Wellcome Trust investigator award 098291/Z/12/Z. MA was funded by Canceropôle PACA (Valo-Paca 2016) and French National Institute of Cancer (Inca, PRT-K16, #2017-24). PC and VR were funded by BBSRC (BB/M006174/1) and the Barts and The London Charity (297/2249). IJW was funded by an MRC LMCB core grant award MC_U12266B. The funders had no role in study design, data collection and analysis, decision to publish, or preparation of the manuscript.

**Competing interests:** The authors have declared that no competing interests exist.

**Abbreviations:** ACN, acetonitrile; APEX-2, ascorbate peroxidase 2; CBL, Casitas B-lineage lymphoma; CD, cluster of differentiation; Cdc42, cell division control protein 42; DAB, diaminobenzidine; DUB, deubiquitinating enzyme; EGFP, enhanced green fluorescent protein; EGFR, epidermal growth factor receptor; EM, electron microscopy; ESAM, endothelial cell-selective adhesion molecule; ESCRT, endosomal sorting complex required for transport; GFP, green fluorescent protein; HA, hemagglutinin; HGM, HUVEC growth medium; HRP, horseradish peroxidase; Hrs, hepatocyte growth factor-regulated tyrosine kinase substrate; HUVEC, human umbilical vein endothelial cell; ICAM-1/2, intercellular adhesion molecule 1/2; IL-1, interleukin-1; IP, Immunoprecipitated; Jak-1, janus kinase 1; JAM-C, junctional adhesion molecule-C; KIF, kinesin-like protein; LBRC, lateral border recycling compartment; LDL, low-density lipoprotein; MS/MS, tandem mass spectrometry; MVB, multivesicular body; NRP, neuropilin; Par, partitioning defective protein; PDZ, postsynaptic density protein/drosophila disc large tumour suppressor/zonula occludens protein 1; PECAM-1, platelet endothelial cell adhesion molecule; PEG, polyethylene glycol; PEI, polyethylenimine; PLVAP, plasmalemma vesicle-associated protein; PTPR, receptor-type tyrosine phosphatase; Quad-K, 4 lysine residues mutated to arginine residues; Rap-1, ras-related protein; SEM, standard error of the mean; siRNA, small interfering RNA; SNAP23, synaptosomal-associated protein 23; SNARE, soluble N-ethylmaleimide sensitive factor attachment protein receptor; STAM, signal

Essential to the establishment of the endothelial monolayer are the protein complexes between adjacent cells that create junctions. Junctions are dynamic and are remodelled to control processes such as cell permeability [1] and cell migration [5], and a subset of junctional proteins also control leukocyte transmigration [2]. This subset includes molecules such as vascular endothelial cadherin (VE-Cadherin), platelet endothelial cell adhesion molecule (PECAM-1) (cluster of differentiation [CD]31), CD99, endothelial cell-selective adhesion molecule (ESAM), intercellular adhesion molecule 1 and 2 (ICAM-1 and -2), and the junctional adhesion molecule (JAM) family of proteins (JAM-A, B, and C) [2,6]. These proteins engage in homophilic and heterophilic interactions with neighbouring cells or with transmigrating leukocytes. Control of these interactions determines the stability of the junctions and/or provides unligated receptor for the movement of transmigrating leukocytes [2,7]. For some receptors that are directly required for leukocyte transmigration (such as JAM-A and PECAM-1), this is a well-characterised process regulated by a combination of phosphorylation, proteolytic cleavage, and intracellular trafficking. However, for other receptors such as JAM-C, almost nothing is known about the molecular control of receptor trafficking and its importance to function.

JAM-C is a type 1 integral membrane protein of the immunoglobulin superfamily that mediates numerous endothelial cell functions such as leukocyte transendothelial cell migration [8–12], angiogenesis, and vascular permeability [13–16]. However, there is a marked difference in the function of JAM-C compared to other endothelial junctional receptors. Rather than directly inhibiting leukocyte transmigration, inhibition of JAM-C by genetic means or using blocking antibodies changes the mode of transmigration [8,11,17,18]. In JAM-C–deficient endothelial cells, leukocytes breach the endothelial barrier and then reverse, transmigrating back into the vessel lumen and reducing the overall efficiency of leukocyte traffic. Further, unlike JAM-A, JAM-C surface expression favours changes in permeability following stimulation with thrombin, vascular endothelial growth factor (VEGF), or histamine by inhibiting the activation of the small GTPase ras-related protein 1 (Rap-1) and by activating actomyosin function [15]. Given these functional differences, it is likely for the regulation of JAM-C to be similarly diverse. Together, the broad functional roles of JAM-C in inflammation and vascular biology are illustrated through its involvement in multiple inflammatory disease states such as arthritis [19], peritonitis [9], acute pancreatitis [18,20], ischemia reperfusion injury [10,11], pulmonary inflammation [21,22], and atherosclerosis [23–25].

Whilst little is known about the molecular mechanisms of JAM-C trafficking, current evidence indicates an important role for such a phenomenon in regulating JAM-C function. Firstly, in cultured macro- and microvascular endothelial cells, JAM-C can be redistributed in a stimulus-dependent manner [13,15,26]: activation of endothelial cells with thrombin, VEGF, or histamine increases localisation of JAM-C at the cell surface. Secondly, analysis of inflamed murine tissues by electron microscopy (EM) showed labelling of the extracellular domain of JAM-C within vesicles, at the cell surface, and at junctional regions of endothelial cells. Importantly, distribution of JAM-C was altered within these sites following an inflammatory stimulus (ischemia/reperfusion injury), resulting in a change in the levels of vesicular protein versus junctional and cell-surface protein [10,11]. Finally, enhanced levels of JAM-C have been observed in atherosclerosis and rheumatoid arthritis, some of which likely reflect changes in intracellular trafficking [19,24].

Collectively, current evidence suggests a causal link between redistribution of intracellular JAM-C and physiological and pathological processes involving opening of endothelial cell junctions. To directly address this hypothesis, we investigated the mechanisms underlying JAM-C redistribution and trafficking using wild-type (WT) and mutant variants of this receptor and a combination of light and EM. Furthermore, to identify cotrafficked cell-surface

transducing adapter molecule 1; Tie-2, tyrosine-protein kinase receptor 2; TNF, tumour necrosis factor; TSG101, tumour susceptibility gene 101 protein; VEGF, vascular endothelial growth factor; VEGFR, VEGF receptor; USP, ubiquitin carboxyl-terminal hydrolase; VE-Cadherin, vascular endothelial cadherin; VPS, vacuolar protein sorting-associated protein; WASF2, Wiskott–Aldrich syndrome protein family member 2; WT, wild type.

receptors and trafficking machinery involved in JAM-C internalisation and subcellular localisation, we developed novel, to our knowledge, proximity-labelling mass spectrometry approaches. By perturbing the newly identified pathways of JAM-C redistribution and using mutant variants of this receptor, we show that a key point of regulation is the ubiquitylation of JAM-C by the E3 ligase Casitas B-lineage lymphoma (CBL), which regulates the rate of trafficking and lysosomal degradation.

## Results

### JAM-C is dynamically trafficked from the cell surface

To characterise the amount and localisation of intracellular JAM-C present at steady state in cultured endothelial cells, we carried out immunofluorescence analysis in human umbilical vein endothelial cells (HUVECs) (Fig 1A–1D). In confluent monolayers, JAM-C is primarily localised to the junctions, although punctae of JAM-C are also present in just over half of the cells (Fig 1D). Incubating cells with the vacuolar-type H+-ATPase inhibitor bafilomycin (100 nM for 4 h) (Fig 1C), a reagent that blocks the acidification step essential for lysosomal degradation, markedly increased the number of JAM-C–positive intracellular punctae. These results indicate that JAM-C is constitutively trafficked into vesicles, and at least a proportion of the receptor in resting conditions is targeted for degradation in lysosomes.

We next determined the ultrastructure of intracellular JAM-C pools using EM. We generated a version of JAM-C fused to horseradish peroxidase (HRP) in the membrane-proximal region of the extracellular domain (JAM-C–HRPout). Tagging of JAM-C at this point has previously been shown to have no effect on ligand binding or localisation [27]. We utilised murine JAM-C for expression studies in HUVECs because this allows the depletion of endogenous human JAM-C and rescue with the small interfering RNA (siRNA)-resistant murine orthologue. To determine whether such modification can affect the turnover of JAM-C, we utilised an inhibitor of protein synthesis, cycloheximide. We transfected HUVECs with JAM-C–HRPout or a green fluorescent protein (GFP)-tagged equivalent (JAM-C–GFPout), and 24 h later, we added the inhibitor and monitored JAM-C protein levels by western blot (S1A and S1B Fig). There was no significant difference in turnover of any of the tagged proteins compared to endogenous proteins. Cells expressing JAM-C–HRPout were fixed and labelled with diaminobenzidine (DAB) before analysis by EM. DAB labelling was present in small endocytic clathrin-negative structures budding off from the cell surface (Fig 1F and 1Fi), endosomes (Fig 1F), at the junctions (Fig 1G), and in late endosomes/multivesicular bodies (MVBs) (Fig 1H and 1I).

To determine the kinetics of endogenous JAM-C internalisation, we fed anti-JAM-C antibody and monitored the fluorescent intensity at the junctions over a 2-h period (Fig 2A and 2B). This approach showed that approximately two-thirds of surface protein are removed from the junction within 2 h (Fig 2B), indicating that JAM-C is rapidly removed from the junctions at steady state. Antibody cross-linking has been shown to increase the rate of receptor internalisation [28,29], so the kinetics we characterised potentially represent faster-than-average traffic. The assay does, however, confirm that endogenous JAM-C is trafficked from the surface to intracellular pools.

Endothelial cell junctions are remodelled during vascular growth, angiogenesis, and inflammation, allowing cells to move within the monolayer and leukocytes to cross the endothelial barrier. Once these processes are complete, it is essential for the junctions to reform to once again establish a contiguous barrier. Using immunofluorescence and confocal microscopy to analyse localisation of endogenous JAM-C, we monitored the amount of vesicular traffic associated with conditions that are known to disrupt endothelial barrier function. We first

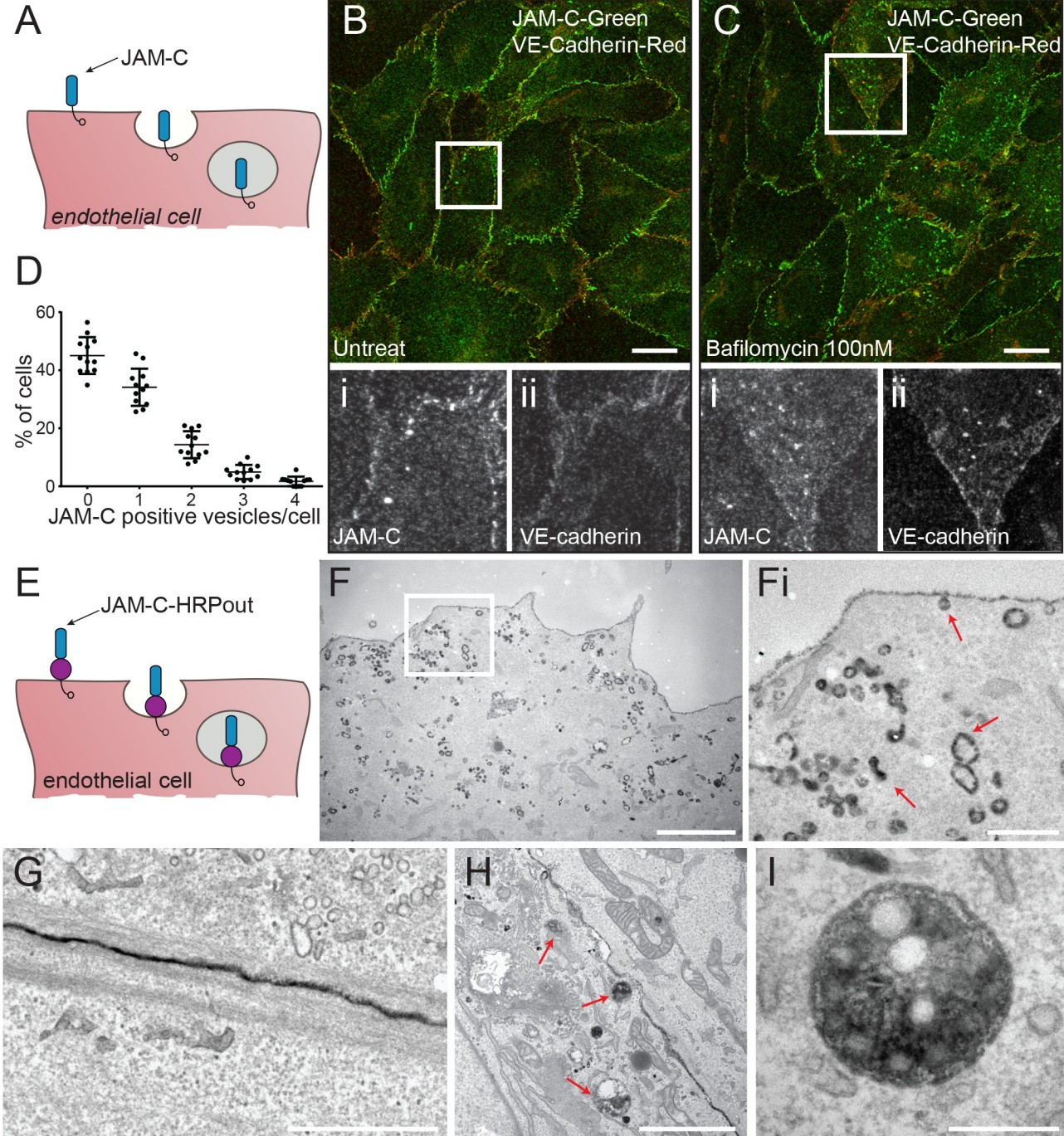

**Fig 1. JAM-C is constitutively trafficked from endothelial cell junctions.** (A) Schematic of the endocytosis of endogenous JAM-C from the endothelial cell surface showing the extracellular immunoglobulin domains (blue) and the intracellular PDZ interacting domain (clear circle). (B) Untreated or (C) 100-nM-bafilomycin–treated HUVECs were fixed and costained for JAM-C (green) and VE-Cadherin (red). Boxed regions are shown magnified below in greyscale: (i) JAM-C and (ii) VE-Cadherin. Scale bar, 20 μm. (D) Quantification of the percent of untreated cells with a number of JAM-C–positive vesicles (551 cells from $n$ = 3 experiments, error bars represent SEM). Underlying data are found in S1 Data. (E) Schematic of the endocytosis of JAM-C–HRPout from the endothelial cell surface showing the extracellular immunoglobulin domains (blue), the HRP tag (purple circle) and the intracellular PDZ interacting domain (clear circle). (F–I) HUVECs were transiently transfected with JAM-C–HRPout, fixed, and incubated with DAB and hydrogen peroxide for 30 min. Cells were then secondarily fixed and 70 nm sections prepared and imaged by transmission EM. Precipitated DAB can be clearly seen on (F) endocytic structures and early endosomes. (Fi) shows boxed area magnified, and red arrows highlight endocytic structures and endosomes: (G) junctions, (H and I) MVBs (red arrows). Scale bars, (F) 2 μm, (Fi) 500 nm, (G) 1 μm, (H) 2 μm, and (I) 200 nm. DAB, diaminobenzidine; EM, electron microscopy; HRP, horseradish peroxidase; HUVEC, human umbilical vein endothelial cell; JAM-C, junctional adhesion molecule-C; MVB, multivesicular body; PDZ, postsynaptic density protein/drosophila disc large tumour suppressor/zonula occludens protein 1; SEM, standard error of the mean; VE-Cadherin, vascular endothelial cadherin.

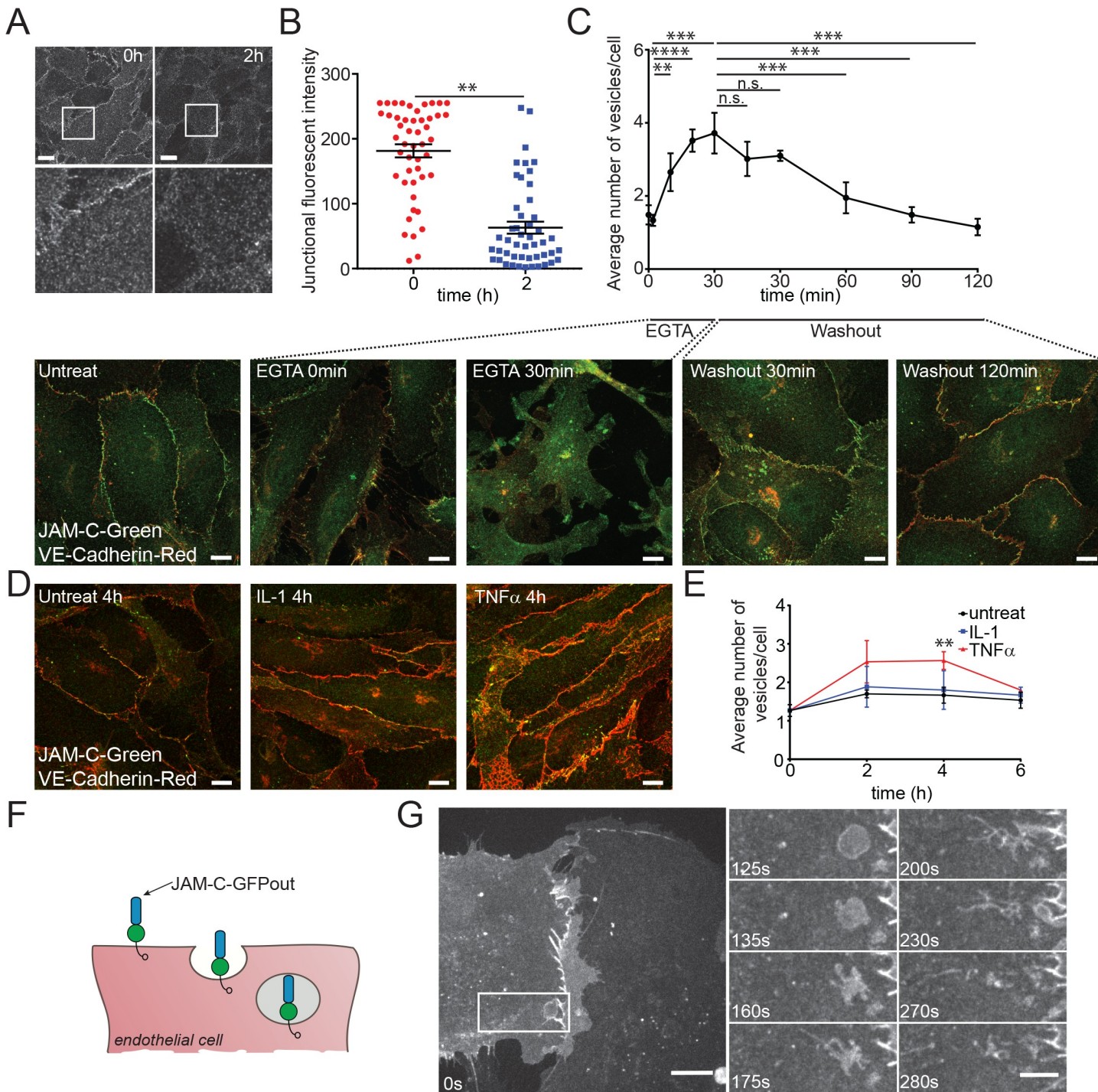

**Fig 2. JAM-C is turned over dynamically at the endothelial junction, and the rate of internalisation is increased during junctional disassembly and inflammation.**
(A) HUVECs were fed with JAM-C antibody, excess antibody was removed, and the cells were incubated at 37°C before being fixed at 0 and 2 h and imaged by confocal microscopy. Scale bar, 20 μm; boxed regions are shown at higher magnification. (B) The difference in junctional intensity is quantified between 0 and 2 h ($n = 4$ experiments, error bars represent SEM, $**p \leq 0.01$; $t$ test). (C) HUVECs were treated with 4 mM EGTA in calcium-free, low-serum media for 30 min before washing and incubation at 37°C with conventional media. Cells were fixed before and during the time course of washout and recovery and stained for JAM-C (green) and VE-Cadherin (red). Images were acquired by confocal microscopy and the number of JAM-C–positive vesicles at each time point determined. Scale bar, 10 μm (5 fields of view/experiment, $n = 3$ experiments, error bars represent SD, $**p \leq 0.01$, $***p \leq 0.001$, $****p \leq 0.0001$; $t$ test). (D) HUVECs were either left untreated or incubated with 10 ng/ml IL-1 or 50 ng/ml TNF-α for 4 h at 37°C. Cells were then fixed and stained for JAM-C (green) and VE-Cadherin (red). Images were acquired by confocal microscopy and (E) the number of JAM-C–positive vesicles at each time point determined. Scale bar, 10 μm (150 cells from $n = 3$ experiments, error bars represent SD, $**p \leq 0.01$; $t$ test). (F) Schematic of the endocytosis of JAM-C–GFPout from the endothelial cell surface showing the extracellular immunoglobulin domains (blue), the

GFP tag (green circle), and the intracellular PDZ interacting domain (clear circle). (G) HUVECs were nucleofected with WT JAM-C–GFPout and imaged with a spinning-disk confocal microscope. Time indicates total time in media, and the boxed region at 0 s is shown magnified at later time points. JAM-C exists in vesicles, at the cell surface, and at the cell junctions. Large vesicles can be seen that tubulate and migrate away from the junctional region. Scale bar: 0 s, 20 μm; inset, 10 μm. Underlying data are found in S2 Data. GFP, green fluorescent protein; HUVEC, human umbilical vein endothelial cell; IL-1, interleukin-1; JAM-C, junctional adhesion molecule-C; n.s., not significant; PDZ, postsynaptic density protein/drosophila disc large tumour suppressor/zonula occludens protein 1; SEM, standard error of the mean; TNF, tumour necrosis factor; VE-Cadherin, vascular endothelial cadherin; WT, wild type.

artificially disrupted junctions by chelating calcium ions essential for junctional formation [30]; we then washed out the chelating agent (EGTA) to monitor junctional reformation (Fig 2C). An increased number of JAM-C–positive vesicles was noted following the addition of EGTA that then reduced following washout and formation of a confluent monolayer (Fig 2C). This indicates that vesicular trafficking of JAM-C is associated with the removal of endothelial cell junctions. We next extended these studies to HUVECs treated with the potent proinflammatory cytokines interleukin-1 (IL-1) and tumour necrosis factor (TNF)-α. TNF-α increased the number of JAM-C–positive intracellular vesicles, whilst IL-1 had little effect (Fig 2D and 2E), suggesting that JAM-C junctional remodelling is stimulus-specific and is associated with treatments that reduce barrier function.

To directly monitor the dynamics of JAM-C trafficking, we transfected HUVECs with a JAM-C–GFPout construct (Fig 2F) and observed the cells by spinning-disk confocal microscopy (Fig 2G, S1 Movie). We noted large membranous structures forming near cellular junctions, from which tubular extensions were seen to emerge that subsequently detached and moved away (presumably along cytoskeletal tracks), dissolving the original structure. Together, the present data demonstrate that JAM-C trafficking is a rapid process, a response that is further increased during junctional remodelling and following activation of endothelial cells with certain inflammatory stimuli.

## Development of a novel, to our knowledge, HRP-based proximity-labelling assay for characterising the trafficking of JAM-C

The endothelial cell junction is comprised of multiple protein complexes, some of which have already been demonstrated to redistribute constitutively and during leukocyte transmigration [31–33]. To define in an unbiased manner the receptors localised with JAM-C at the cell junctions and to determine which of these cotraffic with JAM-C, we developed a JAM-C-HRP–based proximity-labelling protocol [34] (Fig 3A–3C). This approach uses the enzymatic activity of HRP to oxidise fluid-phase–fed biotin tyramide, thus generating biotin phenoxyl radicals that covalently react with electron-rich amino acids (such as Tyr, Trp, His, and Cys) on neighbouring proteins (Fig 3B). The short-lived nature of these radicals results in a small labelling radius (<20 nm) and provides a proximity map of all nearby proteins (Fig 3C). We transfected HUVECs with the JAM-C–HRPout construct (Fig 3A) and fed the cells for 30 min with biotin tyramide. The proteins neighbouring JAM-C were biotinylated during a short (1-min) incubation with hydrogen peroxide, and this reaction was then terminated using molecules that scavenge free radicals. Biotinylated proteins in proximity to JAM-C–HRP were detected at the cell surface and within intracellular vesicles (Fig 3D). Importantly, no biotinylation was detected in the absence of biotin tyramide or hydrogen peroxide (S1C Fig). To specifically identify intracellular stores of JAM-C, we incorporated ascorbate, a membrane-impermeant inhibitor of HRP, into our assay. Ascorbate blocks the proximity-labelling reaction at the cell surface, but not in intracellular stores (Fig 3C). This approach has been previously utilised in EM studies to inhibit cell-surface HRP-catalysed DAB labelling [35,36], but to our knowledge, this is the first time it has been employed for proteomics. We verified that the reaction worked by immunofluorescent labelling of treated cells with fluorescently tagged streptavidin to visualise

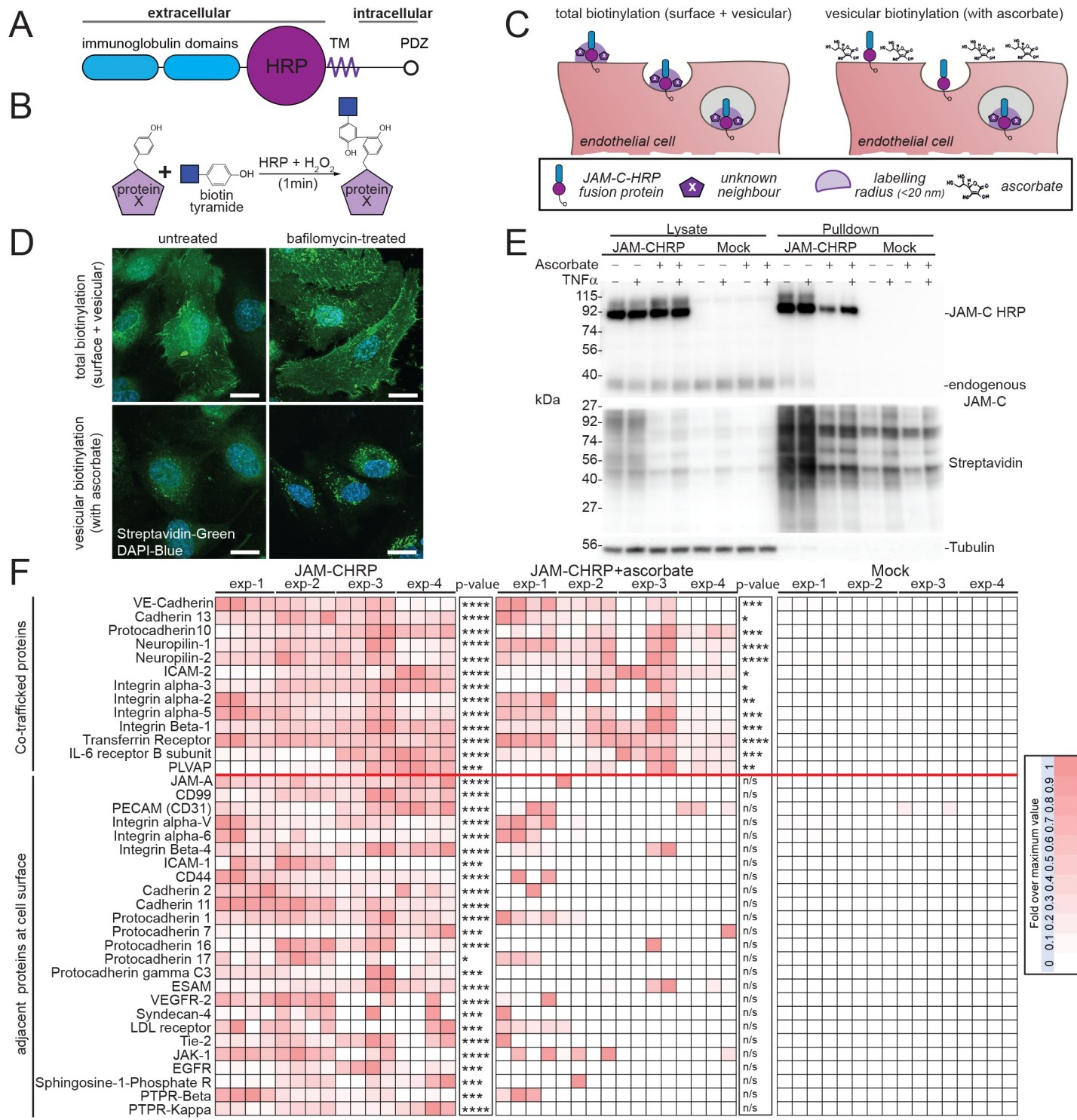

**Fig 3. An HRP-based proximity-labelling approach reveals JAM-C cotraffics with NRP-1 and 2 and VE-Cadherin, but not with components of the LBRC.** (A) Schematic of the domain structure of the JAM-C–HRP construct. (B) In the presence of biotin tyramide and hydrogen peroxide, the HRP tag leads to the biotinylation of any proteins within 20 nm (labelled with an X) on tyrosine, tryptophan, cysteine, or histidine residues. (C) Endothelial cells expressing JAM-C–HRPout are fed with biotin tyramide, and then hydrogen peroxide is added for 1 min. Proteins within a 20-nm radius of JAM-C–HRP are labelled with biotin (shown in light purple). Addition of the membrane-impermeant HRP inhibitor ascorbate blocks the biotinylation reaction at the cell surface but not inside the cell. (D–F) HUVECs were transfected with JAM-C–HRPout and incubated in the presence or absence of bafilomycin A1 (100 nM) for 4 h to block lysosomal degradation. Cells were fed biotin tyramide for 30 min and then exposed to hydrogen peroxide for 1 min in the presence or absence of 50 mM ascorbate. (D) The cells were then fixed and labelled for streptavidin (green) and DAPI (blue) and images acquired by confocal microscopy or (E–F) the reaction was then quenched, and the cells were lysed. Biotinylated

proteins were pulled down using neutravidin beads and lysate, and pulldown samples were analysed by (E) SDS-PAGE and western blot for the presence of JAM-C, tubulin, and biotinylated proteins or (F) following an on-bead tryptic digest mass spectrometry. A heat map of 4 independent mass spectrometry data sets is shown, with white meaning none and dark red a high signal. Individual experiments were carried out in duplicate, with each mass spectrometry run being repeated twice (to give a total of 4 analyses/experiment). *p*-Values are given across all 4 experiments (*$p \leq 0.05$, **$p \leq 0.01$, ***$p \leq 0.001$, ****$p \leq 0.0001$, *t* test). The + TNF-α experiment is shown in S3 Fig. Cotrafficked proteins appear in both ±ascorbate conditions, whilst proteins adjacent to JAM-C solely at the cell surface are only present in the −ascorbate condition. Scale bar, 20 μm. CD, cluster of differentiation; EGFR, epidermal growth factor receptor; ESAM, endothelial cell-selective adhesion molecule; HRP, horseradish peroxidase; HUVEC, human umbilical vein endothelial cell; ICAM, intercellular adhesion molecule; IL, interleukin; Jak-1, janus kinase 1; JAM-C, junctional adhesion molecule-C; LBRC, lateral border recycling compartment; LDL, low-density lipoprotein; NRP, neuropilin; n/s, not significant; PDZ, postsynaptic density protein/drosophila disc large tumour suppressor/zonula occludens protein 1; PECAM, platelet endothelial cell adhesion molecule; PLVAP, plasmalemma vesicle-associated protein; PTPR, receptor-type tyrosine phosphatase; Tie-2, tyrosine-protein kinase receptor 2; TM, transmembrane; TNF, tumour necrosis factor; VE-Cadherin, vascular endothelial cadherin; VEGFR, vascular endothelial growth factor receptor.

biotinylated proteins near JAM-C (Fig 3D and S1C Fig) and by western blotting using streptavidin HRP (Fig 3E). Both methods confirmed that intracellular stores of JAM-C could be visualised, but not cell-surface pools. To further increase the number of intracellular vesicles containing JAM-C, we also incorporated bafilomycin into the assay (Fig 3D). Following the reaction, cells were lysed and biotinylated proteins pulled down with streptavidin before performing an on-bead tryptic digest and subsequent tandem mass spectrometry (MS/MS) analysis.

We detected 134 proteins that were within the vicinity of JAM-C at the cell surface and 50 proteins that were proximal to JAM-C after internalisation; an example of all the data from one replicate is shown, including the background proteins identified (S1D Fig), and a more detailed table showing proteins 3.5-fold enriched over the mock condition and that exhibit differential trafficking is also shown (Fig 3F). Our mass spectrometry results (Fig 3, S1 Table—includes corrected *p*-value for multiple comparisons—and S1E and S1F Fig) indicate that components of a known junctional adhesion molecule trafficking pathway, the lateral border recycling compartment (LBRC), comprising PECAM-1, CD99, and JAM-A [31–33] are all adjacent to JAM-C at the cell surface but are not cotrafficked with JAM-C. Proteins classically associated with endothelial cell permeability such as VE-Cadherin, NRP-1, and NRP-2 were adjacent to JAM-C at the cell surface and also cotrafficked with JAM-C. Other cotrafficked receptors included plasmalemma vesicle-associated protein, a protein implicated in permeability, angiogenesis, and leukocyte transmigration [37], and a number of integrin subunits (alpha-2, 3, and 5 and beta-1). Notably, VEGF receptor (VEGFR)2, a receptor classically associated with the NRPs, is adjacent to JAM-C at the junctions but absent from the intracellular carriers, suggesting that separate trafficking of these coreceptors may be a means of regulating their functions.

We further validated our MS results by immunofluorescence analysis and western blotting of purified biotinylated proteins (S2 Fig). Since TNF-α can increase the number of JAM-C–positive vesicles (Fig 2E), to determine whether these are qualitatively different to those in unstimulated cells (i.e., the result of an additional inflammatory pathway) or whether they simply represent an up-regulation in the rate of traffic, we carried out HRP proximity labelling in the presence of TNF-α. The cotrafficking proteins were largely the same (S3 Fig). The only major difference noted was a potential reduction in the cotrafficking of JAM-C with VE-Cadherin, and this needs to be verified in future work. This indicates that inflammatory stimuli affect the rate, but not the primary route, of JAM-C intracellular traffic.

## JAM-C turnover is dependent on ubiquitylation of the cytoplasmic tail

To determine the importance of intracellular trafficking on the function of JAM-C, we needed to define the mechanism of its turnover. Trafficking of a receptor is most often governed by specific signals and motifs present in the intracellular domain. The cytoplasmic domain of

JAM-C is relatively short (44 amino acids) and features a number of conserved potential phosphorylation and ubiquitylation sites, as well as a postsynaptic density protein/drosophila disc large tumour suppressor/zonula occludens protein 1 (PDZ) interacting domain (Fig 4A). We incorporated mutations of all these sites into our GFPout JAM-C constructs either by site-directed or truncation mutagenesis. When expressed in endothelial cells, all mutants exhibited at least some junctional localisation (Fig 4B). The ΔPDZ mutant exhibited a noticeably different localisation in that it was expressed at higher levels and was also found to be enriched in a peri-Golgi pool. To determine the effect of each mutation on JAM-C-GFP turnover, we transiently transfected HUVECs with the JAM-C constructs and, 16 h later, added cycloheximide to block new protein synthesis. We then monitored the loss of GFP-labelled protein by western blot over a 24-h period (Fig 4C and 4D). We observed that the K283R and Y267A mutants had slower and quicker turnover than WT JAM-C, respectively. Given that increases in cell-surface JAM-C levels are associated with a number of disease states, including atherosclerosis and rheumatoid arthritis [19,24], and the K283R mutant exhibits a slower turnover, we focused on the potential role of ubiquitylation in JAM-C trafficking and degradation.

JAM-C has 4 lysine residues in its cytoplasmic tail (Fig 5A); to completely abrogate ubiquitylation, we needed to generate a mutant with all lysine residues mutated to arginine (Quad-K JAM-C). We co-transfected HUVECs with hemagglutinin (HA)-tagged ubiquitin and either WT or mutant JAM-C–GFPout or endothelial growth factor receptor (EGFR)-GFP (as a positive control). Using GFP trap beads, we pulled down the tagged protein and quantified levels of GFP-tagged receptor and associated HA-tagged ubiquitin (Fig 5B). The Quad-K JAM-C was devoid of ubiquitylation, unlike both the WT and EGFR protein. No obvious change in JAM-C localisation was apparent with the ubiquitin mutant in fixed images (Fig 5C). However, live-cell imaging of WT and Quad-K JAM-C–GFPout in the presence of bafilomycin revealed marked differences in receptor trafficking (Fig 5D and 5E). The WT receptor gradually accrues in vesicles following bafilomycin treatment (Fig 5D, S2 Movie). These structures represent the pool of late endosomes and lysosomes unable to degrade because of the inhibition of V-ATPase function and therefore acidification. In contrast, Quad-K JAM-C–GFPout cycles between endocytic structures and the cell surface and does not accrue in late endosomal/lysosomal structures (Fig 5E, S3 Movie). This demonstrates fundamental differences in receptor trafficking following blocking of JAM-C ubiquitylation. To more closely examine the ultrastructure of the intracellular pools in which the Quad-K JAM-C mutant localises, we incorporated the lysine mutations into the JAM-C–HRPout construct and expressed it in unstimulated endothelial cells before carrying out DAB labelling and EM analysis. We found clear evidence of Quad-K JAM-C–HRP on early endosomes (Fig 5Fa) but little labelling on MVBs (Fig 5Fb and 5Fc), in contrast to WT JAM-C–HRPout (Fig 1H and 1I). These data are consistent with the live-cell imaging experiments and indicate that the mutation of intracellular lysine residues in JAM-C prevents sorting of the receptor onto the intraluminal vesicles of the MVB for subsequent degradation.

### An ascorbate peroxidase 2 proximity-labelling approach to identify machinery utilised for JAM-C trafficking

To determine the molecular machinery associated with the intracellular trafficking and ubiquitylation of JAM-C, we used an ascorbate peroxidase 2 (APEX-2) proximity-labelling approach. This strategy was originally developed by Rhee and colleagues [34]. To specifically identify machinery associated with ubiquitylation of JAM-C, we compared WT JAM-C–APEX-2 biotinylation with the Quad-K JAM-C–APEX-2 and monitored the proteins enriched in the former over the latter.

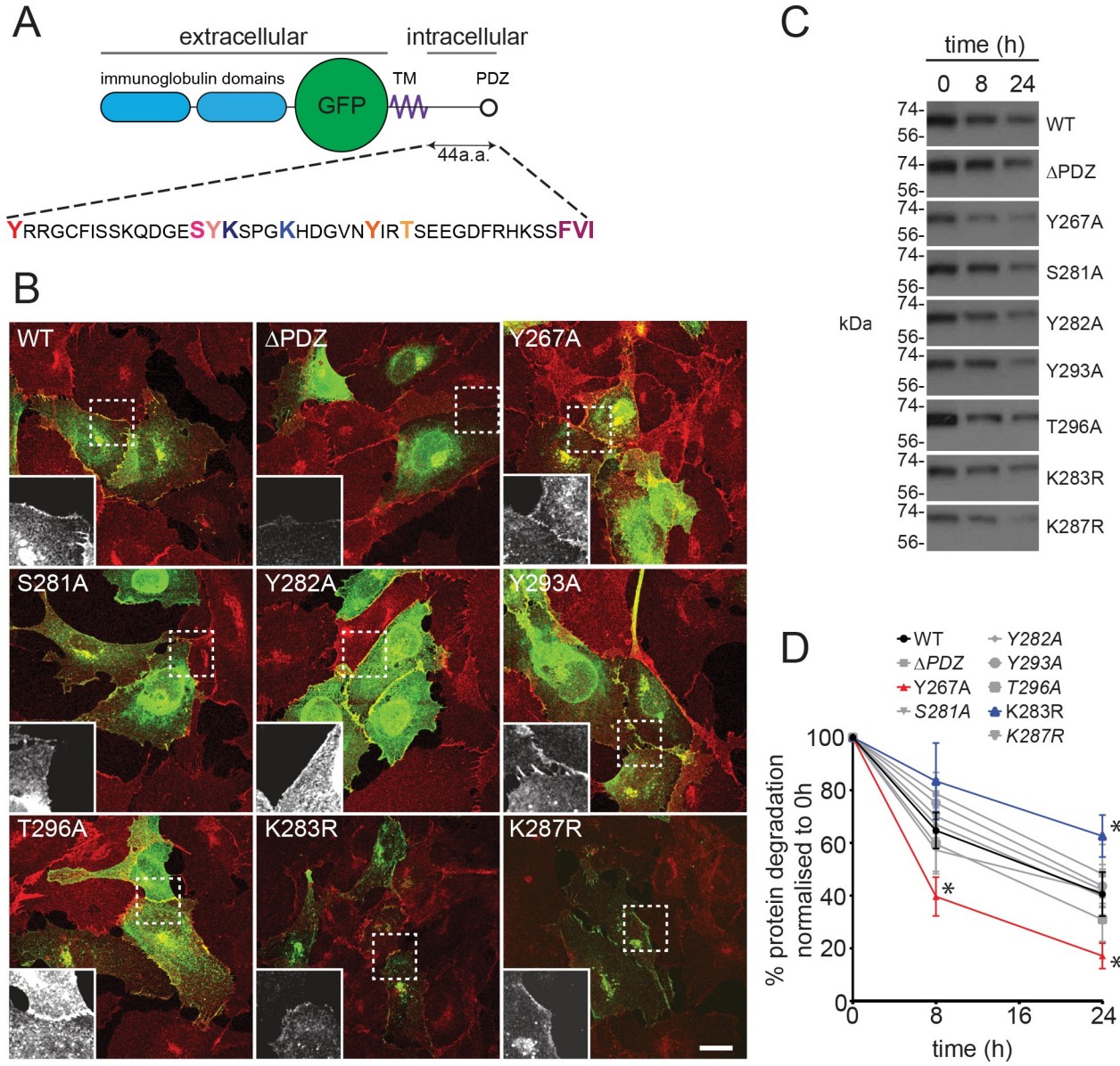

**Fig 4. A lysine residue in the cytoplasmic tail of JAM-C is required for its timely degradation.** (A) A schematic representation of the JAM-C–GFPout construct showing the residues in the cytoplasmic tail that were mutated by site-directed mutagenesis. (B) HUVECs were transfected with WT or mutant JAM-C–GFPout constructs, fixed, and stained for VE-Cadherin (red). The boxed area is magnified (bottom left of each panel) to show the Jam-C–GFP signal. All JAM-C mutants exhibit at least some junctional staining. The ΔPDZ construct exhibited the most marked change in localisation, existing additionally in a peri-Golgi region and expressing at higher levels. Scale bars, 20 μm. (C and D) HUVECs were transfected with WT or mutated JAM-C–GFPout constructs and treated with 10 μg/ml cycloheximide. (C) The rate of protein degradation was determined by SDS-PAGE and western blot over a 24-h period; a representative anti-GFP blot is shown. (D) Quantification of western blots normalised to the 0-h protein levels, data from $n = 7$ experiments, error bars represent SEM (*$p \leq 0.05$; two-way ANOVA). Underlying data are found in S3 Data. a.a., amino acid; GFP, green fluorescent protein; HUVEC, human umbilical vein endothelial cell; JAM-C, junctional adhesion molecule-C; PDZ, postsynaptic density protein/drosophila disc large tumour suppressor/zonula occludens protein 1; SEM, standard error of the mean; TM, transmembrane; WT, wild type.

We incorporated a codon-optimised APEX-2 tag in the membrane-proximal region of the cytoplasmic tail of JAM-C in both WT and Quad-K JAM-C (Fig 6A). This site was chosen so as to minimise the potential for disrupting the C-terminal PDZ interacting domain and other motifs present in the cytoplasmic tail. To ensure the WT and mutant JAM-C–APEX constructs

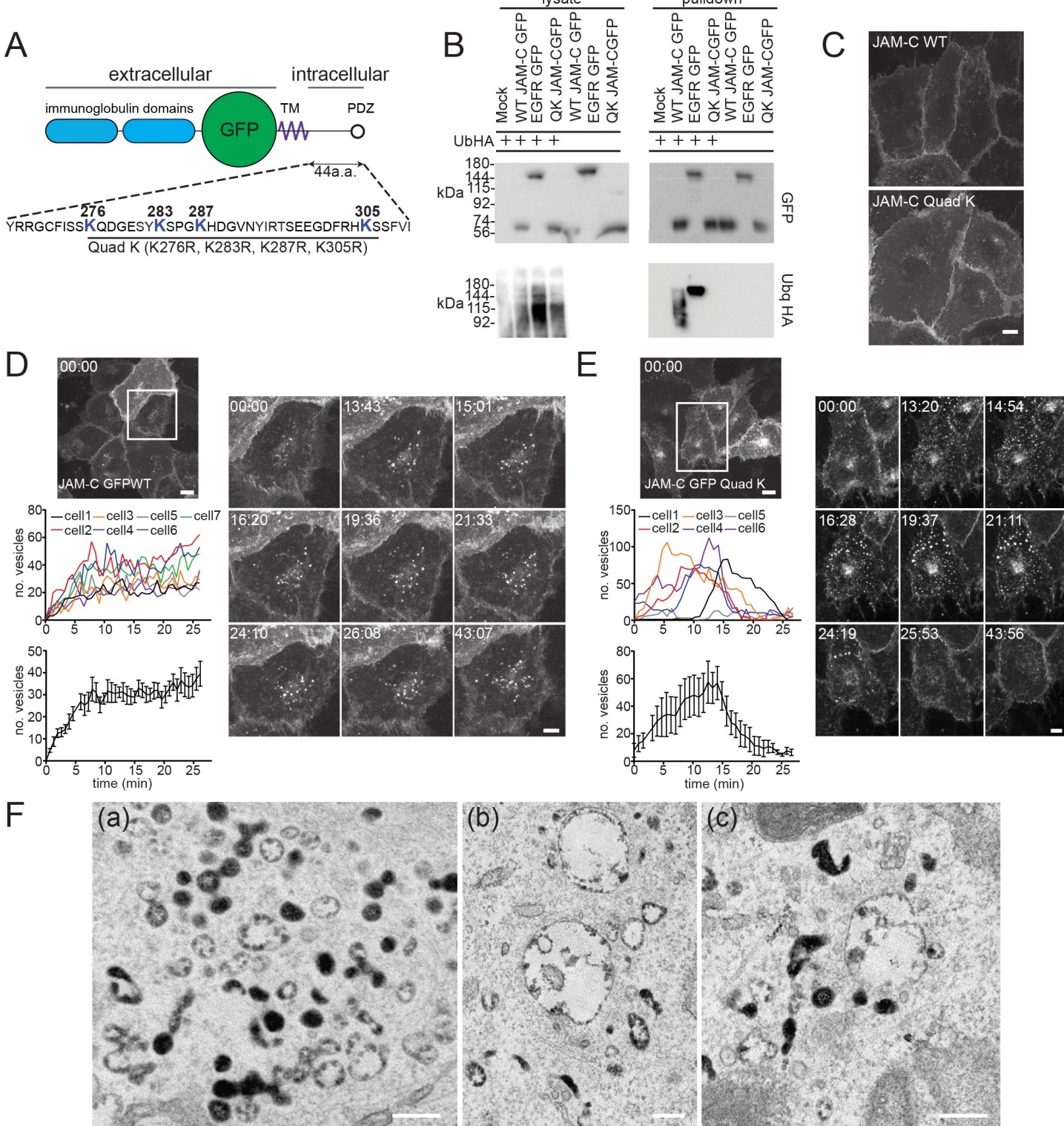

**Fig 5. JAM-C is ubiquitylated, and this governs its targeting to MVBs.** (A) A schematic representation of the JAM-C–EGFPout construct showing all the lysine residues in the cytoplasmic tail that can be potentially modified by ubiquitylation and the JAM-C mutant generated. (B) HUVECs were transfected with WT, Quad-K JAM-C–GFPout, or EGFR-GFP (as a positive control) and ubiquitin HA. Cells were lysed, and GFP-tagged proteins were pulled down using GFP trap agarose beads. Lysate and pulldown proteins were analysed by SDS-PAGE and western blot for the presence of GFP- and HA-tagged proteins. WT, but not Quad-K, JAM-C is modified with ubiquitin. (C) HUVECs were transfected with WT or Quad-K mutant JAM-C–GFPout and then fixed before imaging by confocal microscopy. Scale bar, 20 μm. (D

and E) HUVECs were transfected with (D) WT or (E) Quad-K JAM-C–EGFPout and 100 nM bafilomycin was added (at $t$ = 0 min) before imaging by time-lapse confocal microscopy. The number of vesicles over time in each cell or the average number of vesicles/cell is plotted (error bars represent SEM). A representative movie is shown of $n$ = 3 separate experiments. Boxed regions are shown magnified as individual stills. Scale bars, 20 μm and 5 μm in magnified stills. (F) HUVECs were transiently transfected with Quad-K JAM-C–HRPout, fixed, and incubated with DAB and hydrogen peroxide for 30 min. Cells were then secondarily fixed, and 70-nm sections were prepared and imaged by transmission EM. Precipitated DAB can be clearly seen on early endosomes, but there is little evidence of labelling on MVBs (a, b, and c). Scale bar, 200 nm. Underlying data are found in S4 Data. a.a., amino acid; DAB, diaminobenzidine; EGFP, enhanced green fluorescent protein; EGFR, epidermal growth factor receptor; EM, electron microscopy; GFP, green fluorescent protein; HA, hemagglutinin; HRP, horseradish peroxidase; HUVEC, human umbilical vein endothelial cell; JAM-C, junctional adhesion molecule-C; MVB, multivesicular body; PDZ, postsynaptic density protein/drosophila disc large tumour suppressor/zonula occludens protein 1; QK, Quad-K; Quad-K, 4 lysine residues mutated to arginine residues; TM, transmembrane; Ub/Ubq, ubiquitin; WT, wild type.

did not dimerise with the endogenous WT receptor, we depleted the endogenous receptor and expressed siRNA-resistant constructs (Fig 6B). We detected robust biotinylation by western blot following 30-min feed with biotin tyramide and 1-min exposure to hydrogen peroxide (Fig 6C). Moreover, biotinylated proteins were detected at the junctions and on intracellular vesicles following immunofluorescence labelling with streptavidin 488 (Fig 6D). To identify proteins neighbouring JAM-C–APEX, we pulled down biotinylated proteins and carried out on-bead trypsin digestion and MS/MS analysis. Mass spectrometry analysis revealed 576 significant hits that were more than 3.5-fold enriched over the mock transfected cells (Fig 6E and 6F, S2 Table). Hits included proteins from the cell-surface and intracellular pools and associated with a number of cellular processes, including endocytosis, recycling, fusion, and ubiquitylation. We detected a number of integral membrane proteins previously identified from our HRP approach, including VE-Cadherin. Of particular interest were a number of components associated with endosomal sorting complex required for transport (ESCRT)-0 and 1 and E3 ligases such as CBL, demonstrating that at some point during its trafficking, JAM-C is adjacent to machinery necessary for the ubiquitylation and the maturation of endosomes to MVBs. The equivalent analysis of Quad-K JAM-C–APEX-2 biotinylation showed a similar profile, but notably, CBL and components of the ESCRT complex, vacuolar protein sorting-associated protein (VPS)28 and Vps4B, were absent (Fig 6E and 6G). This indicates that when the 4 lysines present in the cytoplasmic tail of JAM-C are mutated, the E3 ligase and some components of the associated ESCRT machinery are not recruited, and JAM-C is therefore not sorted onto intraluminal vesicles.

To confirm that CBL is required for the ubiquitylation of JAM-C, we reduced its expression using siRNA transfection in HUVECs (Fig 7A). Depletion of CBL decreased the ubiquitylation of JAM-C–GFPout (Fig 7B and 7C) and increased the levels of receptor present in endothelial cells (Fig 7B and 7D). The effects of CBL depletion are particularly apparent when the levels of ubiquitylation are normalised to the levels of JAM-C–GFPout present (Fig 7E). CBL knockdown caused no significant change in the surface localisation of endogenous receptor by immunofluorescence analysis (Fig 7F), although this was confounded by the variability in junctional quantification of immunofluorescence images. However, on addition of bafilomycin, CBL-depleted cells have significantly fewer JAM-CGFPout–positive vesicles (Fig 7F and 7G). This indicates that CBL is required for JAM-C ubiquitylation, which is necessary for JAM-C sorting onto the intraluminal vesicles of late endosomes.

## Ubiquitylation of JAM-C is required for cell migration

JAM-C is known to mediate a number of important biological processes, including angiogenesis [16,38–41] and leukocyte transmigration [8–12]. To determine how altering the ubiquitin-mediated trafficking of JAM-C affects its cellular function, we performed scratch wound assays. Recovery of the wound has been shown to be reduced in conditions of JAM-C knockdown [39] and increased in situations of JAM-C overexpression [42]. This assay provides a

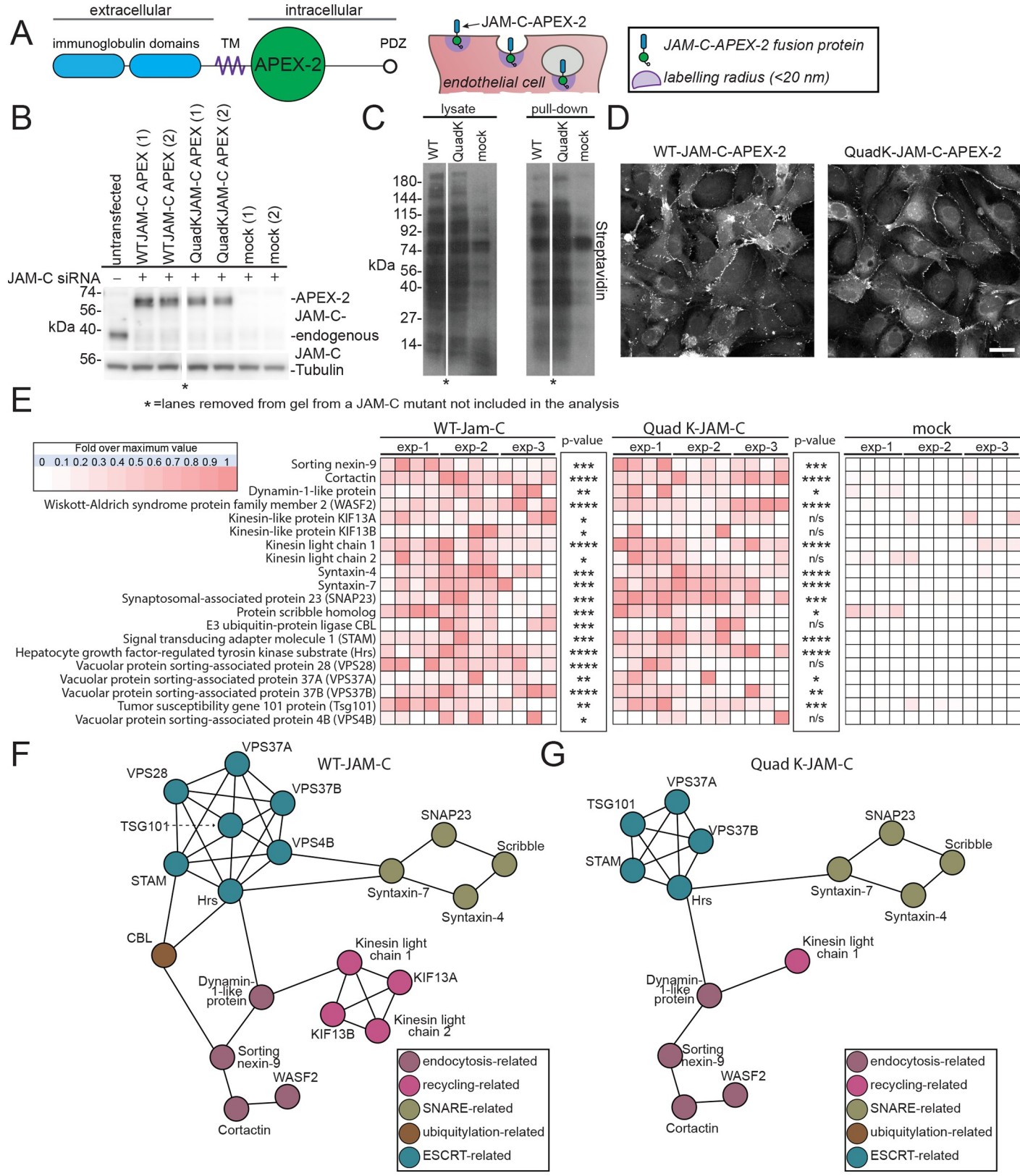

**Fig 6. An APEX-2 proximity-labelling approach reveals potential JAM-C trafficking machinery.** (A) Schematic of the domain structure of the JAM-C–APEX-2 construct, showing the extracellular immunoglobulin domains (blue), the APEX-2 tag (green circle), and the intracellular PDZ interacting domain (clear circle). The

APEX-2 tag, following feeding with biotin tyramide and incubation with hydrogen peroxide for 1 min, labels all neighbouring proteins within 20 nm on electron-rich amino acids (purple circle). (B–E) HUVECs were transfected with WT or Quad-K JAM-C–APEX-2. Cells were fed biotin tyramide for 30 min and then exposed to hydrogen peroxide for 1 min. The reaction was then quenched, and the cells were either (D) fixed and stained with streptavidin and analysed by confocal microscopy or (B, C, and E) lysed and biotinylated proteins pulled down using neutravidin beads for analysis by (B and C) SDS-PAGE and western blot for the presence of (B) JAM-C and tubulin or (C) biotinylated proteins or (E) following an on-bead tryptic digest, mass spectrometry. A heat map of $n = 3$ independent mass spectrometry data sets is shown, with white representing no signal and dark red a high signal. Each individual experiment was carried out in duplicate, with each mass spectrometry run being repeated twice (to give a total of 3 analyses/experiment). $p$-Values are given across all 3 experiments (*$p \leq 0.05$, **$p \leq 0.01$, ***$p \leq 0.001$, ****$p \leq 0.0001$; $t$ test). (F and G) String analysis of proteins neighbouring (F) WT or (G) Quad-K JAM-C–APEX-2 proteins. *Lane removed of a JAM-C mutant not included in this analysis. APEX-2, ascorbate peroxidase 2; CBL, Casitas B-lineage lymphoma; ESCRT, endosomal sorting complex required for transport; Hrs, hepatocyte growth factor-regulated tyrosine kinase substrate; HUVEC, human umbilical vein endothelial cell; JAM-C, junctional adhesion molecule-C; KIF, kinesin-like protein; n/s, not significant; Quad-K, 4 lysine residues mutated to arginine residues; PDZ, postsynaptic density protein/drosophila disc large tumour suppressor/zonula occludens protein 1; siRNA, small interfering RNA; SNAP23, synaptosomal-associated protein 23; SNARE, soluble N-ethylmaleimide sensitive factor attachment protein receptor; STAM, signal transducing adapter molecule 1; TM, transmembrane; Tsg101, tumour susceptibility gene 101 protein; VPS, vacuolar protein sorting-associated protein; WASF2, Wiskott–Aldrich syndrome protein family member 2; WT, wild type.

simple way to monitor the migration of endothelial cells, a process that is essential during angiogenesis. Such changes are unlikely to be due to altered proliferation because no change in cell growth has been detected following JAM-C overexpression over a 96-h period [43]. We depleted endogenous JAM-C, rescued it with lentiviral-expressed GFP, WT JAM-C–GFPout, or Quad-K JAM-C–GFPout (Fig 8A), and monitored the rate of scratch wound closure over a 16-h period (Fig 8B). The efficiency of endogenous JAM-C knockdown was similar in all conditions, and the expression levels of the JAM-C constructs matched endogenous levels (Fig 8A). Knockdown of JAM-C resulted in a slower rate of scratch wound closure, and this was rescued by overexpression of JAM-C–GFPout (Fig 8B and 8C). In contrast, Quad-K JAM-C–GFPout failed to rescue the phenotype (Fig 8B and 8C), and 60 h postscratch, the wound remained open (Fig 8D).

Finally, we used live-cell imaging to directly analyse the trafficking of JAM-C in migrating cells (Fig 9A, S4 Movie). We noted large JAM-C-GFPout–positive vesicles forming at junctional regions, particularly on either side of the cell's leading edge; these sites are associated with junctional disassembly. Markedly less vesicle traffic was apparent at the very front of migrating cells. This indicates that junctional disassembly is a crucial aspect of cell migration, and a proportion of this internalised JAM-C needs to be degraded by a ubiquitin-mediated lysosomal pathway to allow normal cell migration (Fig 8). Together, the results demonstrate dynamic JAM-C trafficking and ubiquitylation are essential responses for endothelial cell migration.

## Discussion

Prior to this study, many aspects of the molecular control of JAM-C function were unclear, as was the relationship, in terms of regulation and function, with its homologue JAM-A and other components of the junctional machinery. Our presented work addresses many of these issues and opens further avenues of research to determine how intracellular trafficking controls JAM-C's important physiological and pathological roles.

Our proximity-labelling proteomics approach allowed a bulk analysis of the receptors trafficked alongside JAM-C. With this approach, we discovered that JAM-C traffics with VE-Cadherin and NRP-1 and 2, as well as some integrin subunits, but is not present in the same pool as its homologue JAM-A or any of the other component of the LBRC. This cotrafficking is likely to represent receptors associated with a similar function being moved en masse. Receptors present in the LBRC are thought to provide a reticular pool of unligated receptors necessary for leukocyte movement, whilst by contrast, we hypothesise trafficking of JAM-C and its counterparts allows precise control of cell migration and the passage of molecules and cells by disassembling and reforming the junction. In agreement with this, we see increased trafficking

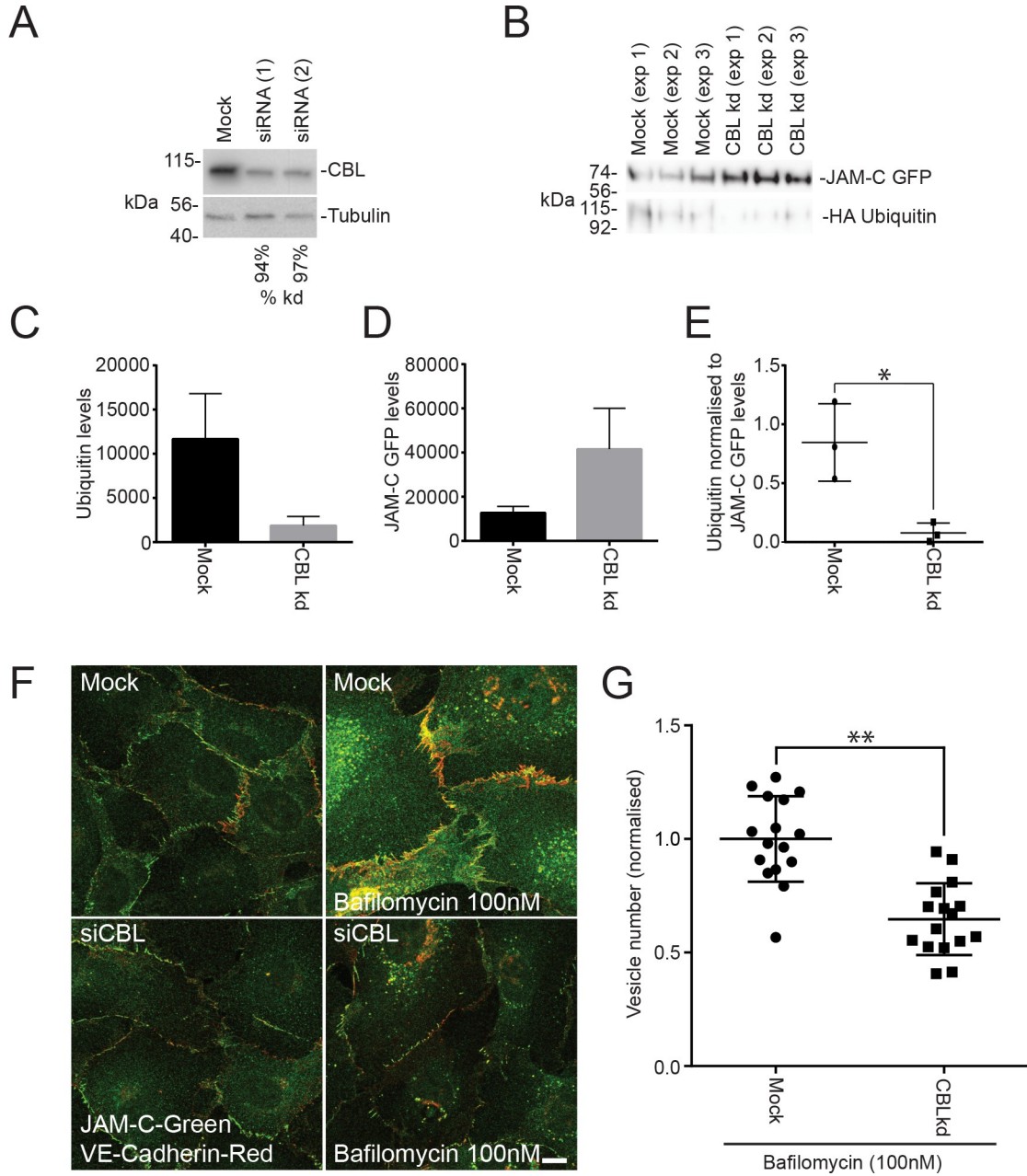

**Fig 7. JAM-C is ubiquitylated by the E3 ligase CBL.** (A–G) HUVECs were transfected with 2 different CBL siRNAs over a 96-h period. (A) Cells were lysed and analysed by SDS-PAGE and western blot, detecting CBL and tubulin. Knockdown was typically ≥90%. (B–E) Untreated or CBL knockdown cells were transfected with JAM-C–GFPout and HA ubiquitin. Cells were lysed, and GFP-tagged proteins were pulled down using GFP trap agarose beads. Pulldown proteins were analysed by SDS-PAGE and western blots for the presence of GFP- and HA-tagged proteins. Quantification of (C) HA ubiquitin levels, (D) JAM-C–GFPout levels, or (E) ubiquitin levels normalised to levels of JAM-C. Results are from $n = 3$ experiments. Error bars represent SEM (*$p \leq 0.05$; $t$ test). (F) Mock and CBL knockdown cells were incubated with and without 100 nM bafilomycin for 4 h and then fixed and labelled for endogenous VE-Cadherin (red) and JAM-C (green). (G) Quantification of the number of JAM-C–positive vesicles present in mock and CBL knockdown cells treated with 100 nM bafilomycin (5 fields of view/condition). Results are shown normalised to the mean number of vesicles in the mock cells from $n = 3$ experiments. Error bars represent SD (**$p \leq 0.01$; $t$ test). Underlying data are found in S5 Data. CBL, Casitas B-lineage lymphoma; GFP, green fluorescent protein; HA, hemagglutinin; HUVEC, human umbilical vein endothelial cell; JAM-C, junctional adhesion molecule-C; kd, knockdown; SEM, standard error of the mean; siRNA, small interfering RNA; VE-Cadherin, vascular endothelial cadherin.

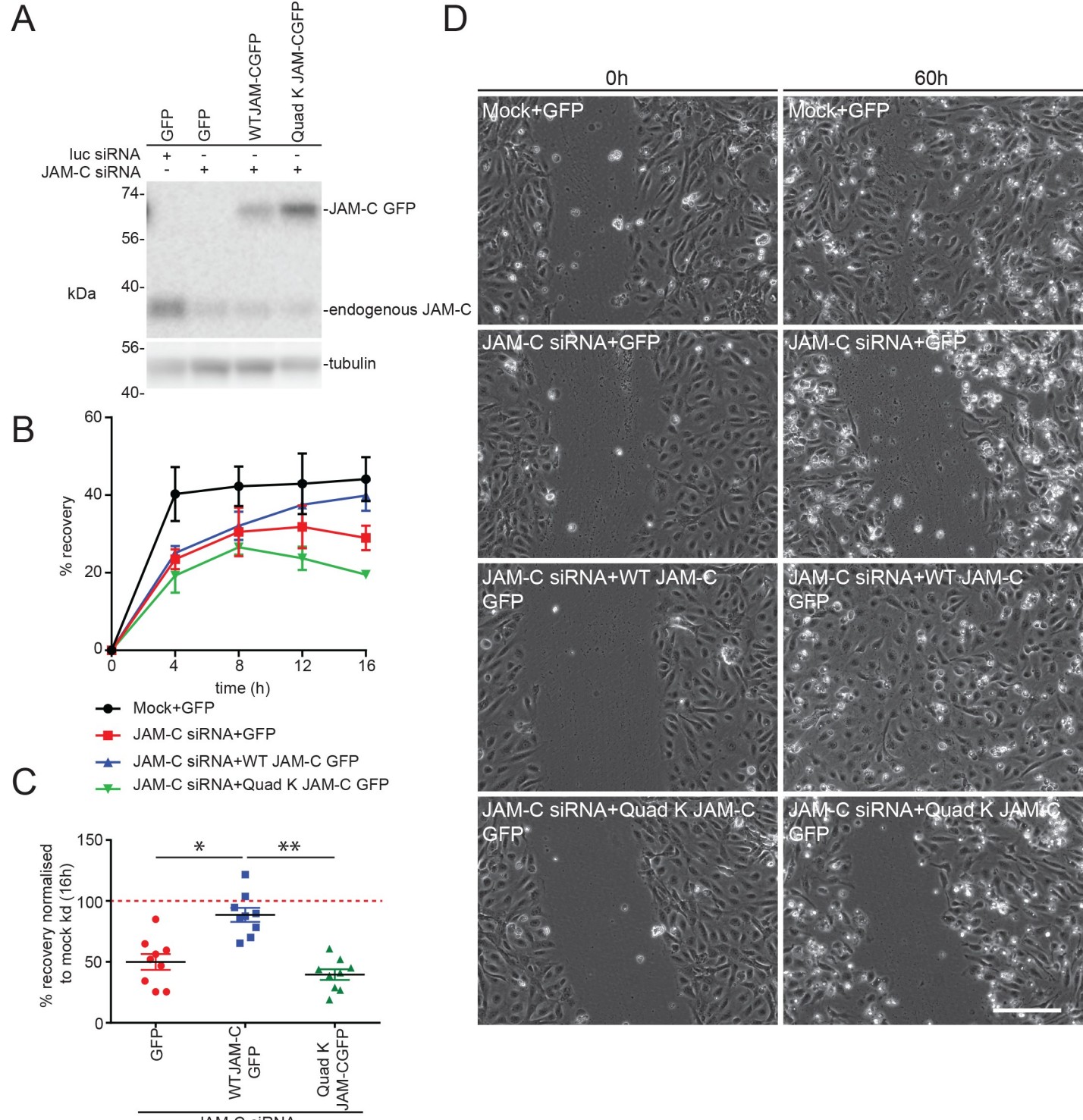

**Fig 8. Ubiquitylation of JAM-C is required for HUVEC migration.** (A–D) HUVECs were transfected with siRNA targeting luciferase or JAM-C over a 96-h period. The day after the second round of transfection, cells were additionally transduced with a GFP-, WT JAM-C-GFPout–, or a Quad-K JAM-C-GFPout–expressing lentivirus and allowed to reach confluency. (A) Cells were lysed, and the extent of JAM-C knockdown and rescue was determined by western blot relative to a tubulin loading control. The expressed WT and Quad-K constructs were similar to endogenous levels. (B–D) The monolayer was scratched, and the cells were allowed to recover over a 0- to 60-h time period whilst being imaged on a brightfield Olympus (Tokyo, Japan) microscope. (B) A representative experiment showing the percentage recovery of cells over a 16-h time period. Mock treated cells expressing the GFP construct recover relatively quickly. JAM-C siRNA-treated cells fail to recover when rescued by transduced Quad-K JAM-C–GFPout or GFP construct. However, almost complete recovery is apparent using transduced WT JAM-C–

GFPout construct after 16 h. Error bars represent SD. (C) The percentage recovery of siRNA-treated GFP, WT JAM-C–GFP, or Quad-K JAM-C–GFP transduced cells at 16 h is shown normalised to the mock/GFP rescued condition. Data shown from n = 4 experiments, error bars represent SEM (*$p \leq 0.05$, **$p \leq 0.01$; unpaired t test). (D) Representative images of recovery of scratch wounds after 60 h. Scale bar, 200 μm. Underlying data are found in S6 Data. GFP, green fluorescent protein; HUVEC, human umbilical vein endothelial cell; JAM-C, junctional adhesion molecule-C; kd, knockdown; luc, luciferase; Quad-K, 4 lysine residues mutated to arginine residues; SEM, standard error of the mean; siRNA, small interfering RNA; WT, wild type.

of JAM-C associated with artificial disruption of the junctions by calcium chelation and following exposure to an inflammatory stimulus (TNF-α). Another cytokine, IL-1, had little effect on trafficking, and this difference likely reflects the ability of TNF-α signalling to additionally stimulate an endothelial permeability response [44]. In vivo, TNF-α signalling is known to be important during ischemia reperfusion injury [45], one of the first occasions in which vesicular pools of JAM-C were noted [10].

Although there is no direct codependence for traffic, it remains possible that JAM-C may share some as yet unknown trafficking machinery with its neighbours and be internalised at the same time from the cell surface. However, some of the receptors present in the same cotrafficked pool have been reported to be internalised by different clathrin-dependent and independent routes [46–50]. An alternative explanation is that the shared pool could represent a fusion of early endosomes from diverse endocytic routes to form a late endosomal pool downstream. There is therefore scope for cotrafficked receptors interacting with each other and therefore modifying function, although this remains to be investigated.

A key finding of the study is that we show for the first time, to our knowledge, that JAM-C is ubiquitylated and that this modification represents a key step in JAM-C receptor trafficking. Our APEX-2 proximity-labelling approach allowed us to identify CBL as the E3 ligase responsible for ubiquitylating JAM-C on up to 4 different residues in the cytoplasmic domain, with the number and pattern of bands suggesting polyubiquitylation. This modification is necessary for the efficient sorting of JAM-C onto intraluminal vesicles of the MVB. The trafficking phenotype on CBL depletion was less striking than that of the Quad-K JAM-C mutant and so likely represents either an incomplete block of the ubiquitylation pathway or compensation by another E3 ligase. Ubiquitylation is a means of controlling turnover of claudin-1, 2, 4, 5, 8, and 16 [51]; however, the only JAM family member previously shown to be ubiquitylated is JAM-B on Sertoli cells. The site of ubiquitylation, the E3 ligase involved, and the effect on receptor trafficking have yet to be determined [52]. A further means to regulate JAM-C trafficking and function is likely to be the removal of ubiquitin. There are thought to be more than 100 potential deubiquitinating enzymes (DUBs) in mammalian cells [51], and we identified 6 potential JAM-C–localised DUBs in our APEX-2 screen: ubiquitin carboxyl-terminal hydrolase (USP) 5, 7, 10, 15, and 47 and USPY. Notably, these proteins were adjacent to JAM-C irrespective of whether we utilised the WT or the Quad-K mutant and therefore could potentially represent nearby bystander proteins rather than bona fide interactors. We have yet to verify whether these proteins are important for JAM-C traffic, although interestingly, USP-47 has been shown to regulate E-cadherin deubiquitylation at junctions [53].

Our results show that targeted removal and degradation of JAM-C from junctions is required for endothelial cell migration. As endothelial cells migrate to fill a gap in the monolayer, endocytic disassembly of the junction occurs on either side of the cells' leading edge. As JAM-C enters endosomes, it becomes ubiquitylated and is directed for degradation by the lysosomes. A failure to ubiquitylate has no effect on overall internalisation but results in premature, mistimed, or mistargeted removal of JAM-C from the cell surface. We present 3 potential models (which are not mutually exclusive) of how inhibition of degradation of JAM-C slows migration (Fig 9B). Firstly, the change in receptor targeting might reduce the kinetics or timing of removal or change the localisation of homo- and heterotypic junctional interactions

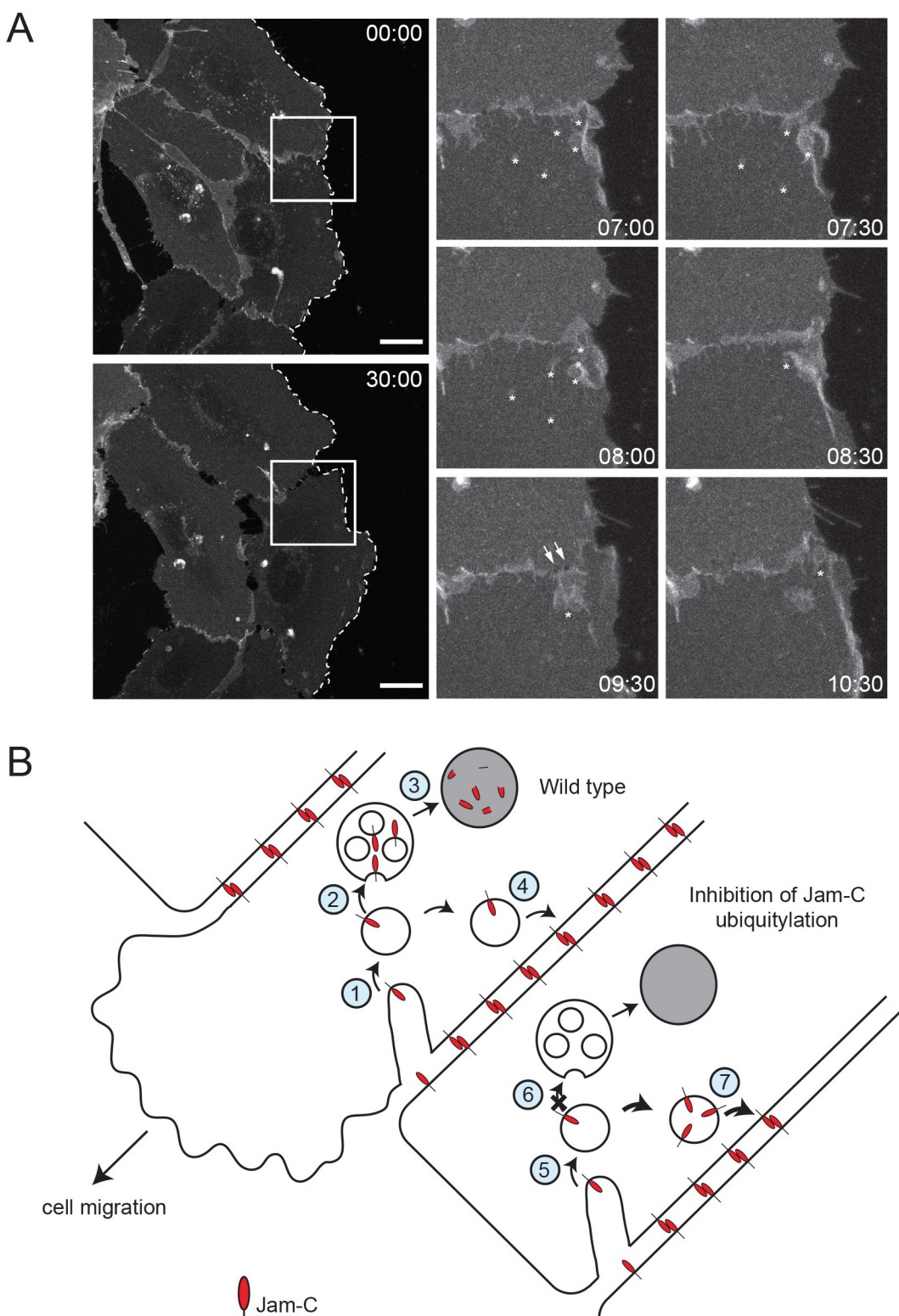

**Fig 9. JAM-C is internalised from junctions near the front of migrating cells.** (A) JAM-C–GFPout was expressed in HUVECs, and the cells were allowed to form a confluent monolayer. The cell layer was scratched, and 45 min later, live-cell imaging was carried out with images acquired every 30 s for 45 min. The dotted line represents the edge of the scratch wound, asterisks highlight vesicles, arrows show holes that form between cells, and the boxed region is shown magnified.

Scale bar, 20 μm. (B) Model of the role of JAM-C trafficking in endothelial cell migration. WT: (1) JAM-C is endocytosed from the junction either side of the leading edge and is localised to endosomes. (2) As JAM-C enters endosomes, the E3 ligase CBL ubiquitylates the cytoplasmic tail on lysine residues, allowing recruitment of the ESCRT complex and the formation of intraluminal vesicles. (3) JAM-C is degraded in the lysosomes, preventing recycling and regulation of integrins and/or abrogating signalling. (4) The remaining JAM-C that is not ubiquitylated is likely returned to the cell surface to reform new junctions after the cell has moved. Inhibition of JAM-C ubiquitylation: (5) JAM-C is endocytosed from the junction either side of the leading edge and is localised to endosomes. (6) Mutation of JAM-C or absence of CBL prevents ubiquitylation and recruitment to the intraluminal vesicles of late endosomes. (7) Failure to degrade JAM-C results in a failure of cell migration potentially due to mistargeting of JAM-C to the appropriate region of the cell junction, misregulation of integrin molecules, and/or abrogation of signalling. CBL, Casitas B-lineage lymphoma; ESCRT, endosomal sorting complex required for transport; GFP, green fluorescent protein; HUVEC, human umbilical vein endothelial cell; JAM-C, junctional adhesion molecule-C; WT, wild type.

with neighbouring cells, preventing cell movement. Secondly, targeting receptor for degradation might be required to disrupt cell-surface protein–protein interactions or terminate signalling events. Such signalling could potentially occur via interactions with the polarity proteins, partitioning defective protein (Par) 3 [54] and Par6 [55]. Notably, Par3/JAM-C interactions are required for neuronal cell migration from the germinal zone [56], and signalling via Par3 and cell division control protein 42 (Cdc42) is required for lumen formation during 3D endothelial tubulogenesis [57]. Thirdly, changes in JAM-C trafficking might alter the ligand binding properties of other receptors. For example, JAM-C has already been shown to regulate integrin [13,42] activation and localisation. If JAM-C is misrouted, there is the potential for this mode of regulation to be changed.

The trafficking and degradation of JAM-C is likely important for the other ascribed roles of JAM-C, particularly during leukocyte transmigration and permeability because they both rely on the disassembly and reassembly of junctions. Controlling the trafficking of JAM-C and its cotrafficked counterparts could potentially provide novel ways of limiting disease states, particularly those associated with increased levels of cell-surface JAM-C such as atherosclerosis and arthritis.

## Methods

### Constructs and cloning

pRK5-HA-Ubiquitin-WT was a gift from Ted Dawson (Addgene plasmid #17608; Cambridge, MA, USA). EGFR-GFP was a gift from Alexander Sorkin (Addgene plasmid #32751). Murine JAM-C–GFPout and JAM-C–GFPout ΔPDZ have been described previously [27]. To generate murine JAM-C–HRPout, HRP was amplified from P-selectin HRP [58] using primers incorporating 15-bp 5′ or 3′ overlaps with JAM-C (S3 Table). JAM-C–GFPout was digested with *Xba*I and *Xho*I and an infusion reaction and subsequent transformation performed according to the manufacturer's instructions (Clontech, Mountain View, CA, USA). QuikChange site-directed mutagenesis of JAM-C–GFPout to generate Y267A, S281A, Y282A, K283R, K287R, Y293A, and T296A was performed according to the manufacturer's instructions (Agilent, Santa Clara, CA, USA) using the primers listed (S3 Table). Murine Quad-K JAM-C–HRPout, JAM-C–APEX-2, and all the site-directed mutations thereof were prepared by gene synthesis (Thermo Fisher Scientific, Waltham, MA, USA). The soybean APEX-2 sequence was first codon-optimised using GeneOptimizer (Thermo Fisher Scientific) and incorporated into the intraluminal domain of JAM-C between the transmembrane and cytoplasmic tail before subcloning into pCDNA3.1 using *Hind*III and *Bam*HI sites. For lentiviral transduction, WT and Quad-K JAM-C–GFPout were amplified using appropriate primers (S3 Table) and subcloned into pLNT-SFFV using an infusion reaction according to the manufacturer's instructions (Clontech).

## Cell culture and transient transfection

HUVECs (Promocell, Heidelberg, Germany) were cultured as previously described [59]. Plasmid transfections were performed by nucleofection (Nucleofector II, programme U-001; Amaxa Biosystems, Gaithersburg, MD, USA) using 2–10 μg DNA or 250 pmol siRNA (JAM-C: AUGUAGUUAACUCCAUCUGGUUUCC, CBL-1: CCUCUCUUCCAAGCAC UGA, CBL-2: CCUGAUCUGACUGGCUUAU, or luciferase: CGUACGCGGAAUACU UCGA).

## Lentivirus preparation

To produce lentivirus, $8 \times 10^6$ HEK293T cells were seeded onto a 15-cm–diameter tissue culture dish in 20 ml DMEM with Glutamax (Gibco, Gaithersburg, MD, USA) with 10% heat-inactivated FBS (Gibco). Cells were incubated overnight before adding 25 μM chloroquine solution 1 h prior to transfection. Transfection complexes were prepared by diluting 18 μg of plasmid expressing the lentiviral packaging genes (pGagPol), 4 μg plasmid expressing the VSV-G envelope (pMDVSVG), and 18 μg of pSFFV-JAM-C-EGFP-WPRE in 2 ml OptiMEM (Gibco). 200 μl polyethylenimine (PEI) solution (1 mg/ml) was added (5:1 ratio of PEI:DNA) and immediately pulse-vortexed (3 brief pulses, low speed so as not to shear DNA). Complexes were incubated at room temperature for 10 min before all volume was transferred to HEK293T cells, which were returned to the incubator overnight before media was aspirated and replaced with 20 ml fresh DMEM for 48 h. Virus-containing supernatant was removed and stored at 4°C and replaced with another 20 ml DMEM complete for a further 24 h. The supernatant was taken and pooled with the earlier material, syringe-filtered with a 0.22-μM pore filter. Four times concentrated polyethylene glycol (PEG) precipitation solution was prepared (36% PEG [6,000 Da], 1.6 M NaCl, filter-sterilised) and diluted to 1× concentrate in the virus-containing supernatant. This was stored at 4°C for 90 min (inverted every 30 min) before centrifugation at $1,500 \times g$ for 1 h at 4°C, following which media was aspirated and the pellet resuspended in OptiMEM. Aliquots were stored at −80°C until use.

## Antibody feeding

HUVECs were cultured on 10-mm coverslips and inverted on prewarmed drops of HUVEC growth medium (HGM) containing a 1:1,000 dilution of rabbit anti-JAM-C antibody (S4 Table) for 15 min. Coverslips were then washed 3 times in prewarmed PBS and transferred to a drop of HGM. Coverslips were fixed at 0, 15, 30 min, 1, 2, or 4 h and immunofluorescence-labelled for total or cell-surface antibody, and the nuclei were labelled with DAPI before analysis by confocal microscopy. Levels of cell-surface JAM-C antibody were determined using Cell Profiler software [60]. Nuclei and edge channels were manually Otsu-thresholded, and a median filter was applied with an artificial diameter of 10 pixels. Objects less than 50 pixels in diameter were filtered out and the object area and intensity determined.

## Ubiquitylation assay

Endogenous CBL was depleted in HUVECs by 2 rounds of transfection with 250 pmol siRNA as above. At the second round of transfection, pRK5-HA-Ubiquitin-WT and WT or mutant JAM-C-GFPout were included; the next day, cells were lysed in RIPA buffer (150 mM NaCl, 1% NP40, 0.5% sodium deoxycholate, 0.1% SDS, and 50 mM Tris [pH 8.0]) supplemented with 10 mM freshly prepared N-ethylmaleimide and protease inhibitors (Sigma-Aldrich, St. Louis, MO, USA). Washed GFP Trap A beads (Chromotek, Planegg-Martinsried, Germany) were added to the lysates and incubated overnight rotating at 4°C. Beads were washed

and resuspended in hot Laemmli buffer (2% SDS, 25% glycerol, 0.36 M β-mercaptoethanol, 0.05 M Tris [pH 8.0]) and western-blotted.

### HRP and APEX-2 proteomics

Typically, 4× 14-cm plates (Nunc, Roskilde, Denmark) of HUVECs were required for each proteomics condition analysed. Endogenous JAM-C was depleted by 2 rounds of transfection over a 96-h period with either 250 pmol JAM-C siRNA/reaction or a nontargeting luciferase siRNA. At the second round, 10 μg WT, Quad-K JAM-C–HRPout, or APEX-2 constructs were also included or, for mock conditions, cells were nucleofected with buffer alone. Following the second round, transfection cells were cultured with 7 μM freshly made haem to aid peroxidase folding and in the presence or absence of 50 ng/ml TNF-α and/or 100 nM bafilomycin (Life Technologies, Carlsbad, CA, USA). 24 h after the second transfection, both mock and peroxidase-transfected cells were fed with 500 μM biotin tyramide (Iris Biotech, Marktredwitz, Germany) for 30 min at 37˚C. The cells were then exposed to M199 supplemented with 1 mM hydrogen peroxide in the presence or absence of 50 mM freshly prepared ascorbate for 1 min. The biotinylation reaction was stopped by the addition of stop solution (PBS, 10 mM sodium azide, 10 mM ascorbate, 5 mM Trolox). Cells were either fixed for immunofluorescence analysis or lysed at 4˚C in RIPA buffer supplemented with 10 mM sodium azide and protease inhibitors. The lysate was centrifuged at $21,000 \times g$ for 15 min at 4˚C and protein concentration determined (Pierce 660 nm Protein Assay Reagent, Thermo Fisher Scientific). Eight point five μg lysate was kept for western blot analysis, and 1.6–1.8 mg total protein was added to 250 μl washed high-capacity neutravidin beads (Life Technologies) in low binding tubes (Life Technologies) and rotated overnight at 4˚C. The beads were washed in 25 mM ammonium bicarbonate buffer, centrifuged at $21,000 \times g$, and frozen at −80˚C before mass spectrometry analysis.

### Cycloheximide chase assay

HUVECs were transfected with WT or mutant JAM-C–GFPout; the next day, cells were washed and incubated with 10 μg/ml cycloheximide. Cells were lysed in RIPA buffer supplemented with protease inhibitors (Sigma-Aldrich) at 0, 8, and 24 h before analysis by western blotting.

### Mass spectrometry

Proteomics experiments were performed using mass spectrometry as reported [61,62]. In brief, Immunoprecipitated (IP) protein complex beads were digested into peptides using trypsin, and peptides were desalted using C18 + carbon top tips (Glygen Corporation, TT2MC18.96; Columbia, MD, USA) and eluted with 70% acetonitrile (ACN) with 0.1% formic acid. Dried peptides were dissolved in 0.1% TFA and analysed by nanoflow ultimate 3000 RSL nano instrument coupled online to a Q-Exactive Plus mass spectrometer (Thermo Fisher Scientific). Gradient elution was from 3% to 35% buffer B in 120 min at a flow rate 250 nL/min, with buffer A being used to balance the mobile phase (buffer A was 0.1% formic acid in water, and B was 0.1% formic acid in ACN). The mass spectrometer was controlled by Xcalibur software (version 4.0) and operated in the positive mode. The spray voltage was 1.95 kV, and the capillary temperature was set to 255˚C. The Q-Exactive Plus was operated in data-dependent mode, with one survey MS scan followed by 15 MS/MS scans. The full scans were acquired in the mass analyser at 375–1,500 m/z with a resolution of 70,000, and the MS/MS scans were obtained with a resolution of 17,500. The mass spectrometry proteomics data have been deposited to the ProteomeXchange Consortium via the PRIDE partner repository with the data set

identifier PXD013003. MS raw files were converted into Mascot Generic Format using Mascot Distiller (version 2.5.1) and searched against the SwissProt database (release December 2015) restricted to human entries using the Mascot search daemon (version 2.5.0) with an FDR of approximately 1%. Allowed mass windows were 10 ppm and 25 mmu for parent and fragment mass to charge values, respectively. Variable modifications included in searches were oxidation of methionine, pyro-glu (N-term), and phosphorylation of serine, threonine, and tyrosine. The Mascot result (DAT) files were extracted into Excel files for further normalisation and statistical analysis.

## Calcium switch

HUVECs were washed in low-calcium media (M199 with 20% dialysed FBS [Sigma-Aldrich] and 10 U/ml Heparin) before addition of low-calcium media with 4 mM EGTA (pH 8.0). After 30 min, the EGTA was removed and replaced with HGM. Cells were fixed before and 30 min after EGTA addition and at 15, 30, 60, and 90 min postwashout. Cells were then analysed by confocal microscopy labelling for JAM-C and VE-Cadherin.

## Live-cell imaging

Endogenous JAM-C or CBL was depleted in HUVECs by 2 rounds of transfection with 250 pmol siRNA as above. At the second round, cells were transfected with WT or Quad-K JAM-C–GFPout and plated on borosilicate glass-bottomed dishes (Greiner Bio One, Kremsmunster, Austria). The following day, cells were imaged in the presence or absence of 100 nM bafilomycin (Life Technologies) in a heat-controlled chamber at 37˚C with 5% $CO_2$ in HGM using either a 63× oil immersion objective (NA 1.3) and a Zeiss 800 microscope (Zeiss, Jena, Germany) or, for higher speed images, a 100× oil immersion lens (NA 1.4) and a spinning disk (UltraVIEW VoX; Perkin-Elmer, Waltham, MA, USA). During longer-term time lapses (45 min), images were acquired every 45 s at a resolution of 512 × 512 pixels and a step size of 0.5 μm. During spinning-disk acquisition, images were acquired every 5 s for 10 min with a step size of 0.4–0.5 μm, comprising 9–14 pictures (depending on cell height) with an exposure for each image at 30 ms.

## Immunofluorescence staining

Fixation and staining were carried out in permeabilised cells as in Lui-Roberts and colleagues [63] using appropriate antibodies (S4 Table). Fixed cell images were taken on a Zeiss 800 scanning confocal microscope system with a 63× objective (NA 1.3) as confocal z-stacks with 0.5-μm step size. Acquisition was performed using Zen Blue software with a 1,024 × 1,024 pixel resolution, 2× frame average, and 1× zoom.

## Western blotting

Proteins were separated by SDS-PAGE, transferred to polyvinylidene fluoride membranes (Perkin-Elmer), and then probed with primary antibody (S4 Table) followed by the appropriate HRP-conjugated secondary antibody (1:5,000) (Agilent).

## Scratch wound assay

Mock or JAM-C knockdown cells were transduced with GFP, WT JAM-C–GFPout, or Quad-K JAM-C–GFPout lentivirus. Scratch wound migration assays were performed on confluent monolayers of HUVECs. Wounds were made using a pipette tip, and images were captured every 30 min for 16 h by time-lapse microscopy using an Olympus (Shinjuku, Tokyo, Japan)

IX81 microscope; additional images of closed scratch wounds were also acquired at 60 h. Percentage wound closure was calculated using Fiji [64].

## EM

HUVECs were transfected with WT or Quad-K JAM-C–HRPout constructs and plated onto glass coverslips. The coverslips were fixed in EM-grade 2% paraformaldehyde, 1.5% glutaraldehyde (TAAB Laboratories Equipment, Ltd., Aldermaston, UK) and washed in 0.05 M Tris HCl (pH 7.6) before incubation with 0.075% DAB, 0.02% hydrogen peroxide for 30 min in the dark. The coverslips were washed in 0.1 M sodium cacodylate and secondarily fixed in 1% osmium tetroxide, 1.5% potassium ferricyanide and then treated with 1% tannic acid. Samples were then dehydrated and embedded in Epon resin. Coverslips were inverted onto prepolymerised Epon stubs and polymerised by baking at 60˚C overnight. Seventy-nm–thick sections were cut with a diatome 45˚ diamond knife using an ultramicrotome (UC7; Leica, Wetzlar, Germany). Sections were collected on $1 \times 2$ mm Formvar-coated slot grids and stained with Reynolds lead citrate. Samples were imaged using a transmission electron microscope (Tecnai G2 Spirit; FEI, Thermo Fisher Scientific) and a charge-coupled device camera (SIS Morada; Olympus).

## Supporting information

**S1 Fig. The JAM-C–HRP biotinylation assay.** (A and B) HUVECs were transfected with WT JAM-C–HRPout or JAM-C–GFPout constructs and treated with 10 μg/ml cycloheximide. (A) The rate of protein degradation was determined by SDS-PAGE and western blot over a 24-h period, and a representative anti-JAM-C blot is shown. (B) Quantification of western blots normalised to the 0-h protein levels; data from $n = 4$ experiments, error bars represent SEM. Underlying data are found in S7 Data. (C–F) HUVECs were transfected with JAM-C–HRP and on the following day incubated with or without biotin tyramide and hydrogen peroxide in the presence or absence of 50 mM ascorbate. (C) Immunofluorescence analysis of cells labelled for streptavidin (green) and EEA 1 (red). Biotinylated proteins are only present when both hydrogen peroxide and biotin tyramide is present and partially localise with early endosomes. (D–F) Biotinylated proteins were pulled down with streptavidin and analysed by mass spectrometry ($n = 4$ experiments). (D) An example data set from one mass spectrometry experiment is shown, consisting of duplicate samples with each mass spectrometry run repeated. Proteins near JAM-C appear only in transfected cells. Proteins that appear solely in mock or in mock and JAM-C-HRP–transfected sample represent nonspecific binders. (E) Pie chart showing the number of proteins adjacent to JAM-C at the cell surface and intracellularly. (F) The percentage of protein hits associated with specific cellular locations and processes is plotted. EEA 1, early endosome antigen 1; GFP, green fluorescent protein; HRP, horseradish peroxidase; HUVEC, human umbilical vein endothelial cell; JAM-C, junctional adhesion molecule-C; SEM, standard error of the mean; WT, wild type.
(TIF)

**S2 Fig. Validation of HRP biotinylation assay by western blot and immunofluorescence analysis.** (A and B) JAM-C-HRPout–transfected cells were fed with biotin tyramide and exposed to hydrogen peroxide in the presence or absence of ascorbate. Biotinylated proteins were pulled down and western-blotted for (A) proteins neighbouring JAM-C at the cell surface: JAM-A or (B) proteins cotrafficked with JAM-C: VE-Cadherin, NRP-1, and NRP-2. Representative blots are shown with quantification of $n = 4$ experiments, and error bars represent SEM ($^{*}p \leq 0.05$, $^{**}p \leq 0.01$; $^{***}p \leq 0.001$, $^{****}p \leq 0.0001$; unpaired $t$ test). (C) Immunofluorescence analysis of endogenous JAM-C (green) and either VE-Cadherin or PECAM-1 (magenta). The

boxed region is magnified. VE-Cadherin cotraffics with JAM-C, whilst PECAM-1 does not. Underlying data are found in S8 Data. HRP, horseradish peroxidase; JAM-C, junctional adhesion molecule-C; NRP, neuropilin; PECAM-1, platelet endothelial cell adhesion molecule 1; SEM, standard error of the mean; VE-Cadherin, vascular endothelial cadherin.
(TIF)

**S3 Fig. An HRP-based proximity-labelling approach reveals changes in JAM-C cotrafficking following stimulation with TNF-α.** (A–C) HUVECs were transfected with JAM-C–HRPout and stimulated for 4 h with 50 ng/ml TNF-α. (A) Cells were lysed and analysed by western blot. The level of JAM-C–HRP expression is similar across all transfected samples, and TNF-α stimulation up-regulates the expression of ICAM-1. (B and C) Cells were fed biotin tyramide for 30 min and then exposed to hydrogen peroxide for 1 min in the presence or absence of 50 mM ascorbate. (B) Cells fixed and stained with streptavidin (green), DAPI (blue), and ICAM-1 (grey). Images were acquired by confocal microscopy. Scale bar, 20 μm. (C) Biotinylated proteins were pulled down using neutravidin beads, and pulldown samples were analysed by mass spectrometry. Heat map of 2 independent mass spectrometry data sets is shown with white meaning no signal and dark red a high signal. Each individual experiment was carried out in duplicate, with mass spectrometry runs being repeated twice (to give a total of 4 analyses/experiment). $p$-Values are given across 2 experiments ($^*p \leq 0.05$, $^{**}p \leq 0.01$, $^{***}p \leq 0.001$, $^{****}p \leq 0.0001$; $t$ test). Cotrafficked proteins appear in both ± ascorbate conditions, whilst proteins adjacent to JAM-C solely at the cell surface are only present in the −ascorbate condition. (D and E) HUVECs were stimulated for 4 h with TNF-α fixed and labelled for (D) JAM-C (green) and VE-Cadherin (magenta) or (E) JAM-C (green) and PECAM-1 (magenta). JAM-C does not colocalise with VE-Cadherin or PECAM-1. Scale bar, 20 μm. HRP, horseradish peroxidase; HUVEC, human umbilical vein endothelial cell; ICAM, intercellular adhesion molecule; JAM-C, junctional adhesion molecule-C; PECAM-1, platelet endothelial cell adhesion molecule 1; TNF, tumour necrosis factor; VE-Cadherin, vascular endothelial cadherin.
(TIF)

**S1 Movie. Spinning-disk microscopy of WT JAM-C–GFPout traffic.** HUVECs were nucleofected with WT JAM-C–GFPout and imaged with a spinning-disk confocal microscope. Time indicates total time in media and a maximum intensity projection is shown of all z-stacked images. JAM-C exists in vesicles, at the cell surface, and at the cell junctions. Large vesicles can be seen that tubulate and migrate away from the junctional region. GFP, green fluorescent protein; HUVEC, human umbilical vein endothelial cell; JAM-C, junctional adhesion molecule-C; WT, wild type.
(MP4)

**S2 Movie. Confocal time-lapse microscopy of WT JAM-C–EGFPout in the presence of bafilomycin.** HUVECs were transfected with WT JAM-C–EGFPout and 100 nM bafilomycin was added (at $t$ = 0 min) before imaging by time-lapse confocal microscopy. A maximum intensity projection is shown of all z-stacked images. EGFP, enhanced green fluorescent protein; HUVEC, human umbilical vein endothelial cell; JAM-C, junctional adhesion molecule-C; WT, wild type.
(MOV)

**S3 Movie. Confocal time-lapse microscopy of Quad-K JAM-C–EGFPout in the presence of bafilomycin.** HUVECs were transfected with Quad-K JAM-C–EGFPout and 100 nM bafilomycin was added (at $t$ = 0 min) before imaging by time-lapse confocal microscopy. A maximum intensity projection is shown of all z-stacked images. EGFP, enhanced green fluorescent protein; HUVEC, human umbilical vein endothelial cell; JAM-C, junctional adhesion molecule-C; Quad-K, 4 lysine residues mutated to arginine residues; WT, wild type.
(MOV)

**S4 Movie. JAM-C is internalised from junctions near the front of migrating cells.** WT JAM-C–GFPout was expressed in HUVECs, and the cells were allowed to form a confluent monolayer. The cell layer was scratched, and 45 min later, live-cell imaging was carried out with images acquired every 30 s for 45 min. A maximum intensity projection is shown of all z-stacked images; scale bar, 20 µm. GFP, green fluorescent protein; HUVEC, human umbilical vein endothelial cell; JAM-C, junctional adhesion molecule-C; WT, wild type. (AVI)

**S1 Table. Mass spectrometry hits from WT JAM-C–HRPout proximity-labelling experiments with and without ascorbate and/or TNF-α.** Mass spectrometry data sets from WT JAM-C–HRPout proximity-labelling experiments with and without 50 mM ascorbate and/or 50 ng/ml TNF-α. Individual experiments were carried out in duplicate, with each mass spectrometry run being repeated twice (to give a total of 4 analyses/experiment for the untreated sample and 2 analyses/experiment for TNF-α experiments). $p$-Values are given across all experiments (*$p \leq 0.05$, **$p \leq 0.01$, ***$p \leq 0.001$, ****$p \leq 0.0001$; $t$ test); corrected $p$-values for multiple comparisons are also given. HRP, horseradish peroxidase; JAM-C, junctional adhesion molecule-C; TNF, tumour necrosis factor; WT, wild type. (XLSX)

**S2 Table. Mass spectrometry hits from WT or Quad-K JAM-C–APEX2 proximity-labelling experiments.** Mass spectrometry data sets from WT or Quad-K JAM-C–APEX2 proximity-labelling experiments. Individual experiments were carried out in duplicate, with each mass spectrometry run being repeated twice (to give a total of 3 analyses/experiment for untreated sample). $p$-Values are given across all experiments (*$p \leq 0.05$, **$p \leq 0.01$, ***$p \leq 0.001$, ****$p \leq 0.0001$; $t$ test); corrected $p$-values for multiple comparisons are also given. APEX-2, ascorbate peroxidase 2; JAM-C, junctional adhesion molecule-C; Quad-K, 4 lysine residues mutated to arginine residues; WT, wild type. (XLSX)

**S3 Table. List of all primers used for mutagenesis and infusion cloning.** (DOCX)

**S4 Table. List of all antibodies used with supplier.** (DOCX)

**S1 Data. Values for each data point to create the graph in Fig 1D.** (XLSX)

**S2 Data. Values for each data point to create the graphs in Fig 2.** (XLSX)

**S3 Data. Values for each data point to create the graph in Fig 4.** (XLSX)

**S4 Data. Values for each data point to create the graphs in Fig 5.** (XLSX)

**S5 Data. Values for each data point to create the graphs in Fig 7.** (XLSX)

**S6 Data. Values for each data point to create the graphs in Fig 8.** (XLSX)

**S7 Data. Values for each data point to create the graph in S1B Fig.**
(XLSX)

**S8 Data. Values for each data point to create the graphs in S2 Fig.**
(XLSX)

**S1 Raw Images. Annotated raw images of all blots presented in figures.**
(PDF)

## Acknowledgments

We gratefully acknowledge the support offered by the Centre for Microvascular Research (CMR) Advanced Bioimaging Facility, part of Queen Mary University of London (QMUL)'s Advanced Molecular Imaging Service, for both their expertise and infrastructure.

## Author Contributions

**Conceptualization:** Katja B. Kostelnik, Amy Barker, Christopher Schultz, Sussan Nourshargh, Thomas D. Nightingale.

**Data curation:** Christopher Schultz, Tom P. Mitchell, Vinothini Rajeeve, Pedro Cutillas.

**Formal analysis:** Katja B. Kostelnik, Amy Barker, Christopher Schultz, Tom P. Mitchell, Vinothini Rajeeve, Ian J. White, Michel Aurrand-Lions, Sussan Nourshargh, Pedro Cutillas, Thomas D. Nightingale.

**Funding acquisition:** Thomas D. Nightingale.

**Investigation:** Katja B. Kostelnik, Amy Barker, Tom P. Mitchell, Ian J. White, Thomas D. Nightingale.

**Methodology:** Katja B. Kostelnik, Amy Barker, Christopher Schultz, Vinothini Rajeeve, Thomas D. Nightingale.

**Project administration:** Thomas D. Nightingale.

**Resources:** Pedro Cutillas.

**Software:** Vinothini Rajeeve, Pedro Cutillas.

**Supervision:** Thomas D. Nightingale.

**Validation:** Katja B. Kostelnik, Amy Barker, Christopher Schultz, Tom P. Mitchell, Thomas D. Nightingale.

**Visualization:** Katja B. Kostelnik, Amy Barker, Christopher Schultz, Ian J. White, Thomas D. Nightingale.

**Writing – original draft:** Thomas D. Nightingale.

**Writing – review & editing:** Katja B. Kostelnik, Amy Barker, Christopher Schultz, Tom P. Mitchell, Vinothini Rajeeve, Ian J. White, Michel Aurrand-Lions, Sussan Nourshargh, Pedro Cutillas, Thomas D. Nightingale.

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
