## [Editor Report · Decision Letter 0]

19 Jul 2019

Dear Dr Nightingale, 

Thank you for submitting your manuscript entitled "Dynamic trafficking and turnover of Jam-C is essential for endothelial cell migration" for consideration as a Research Article by PLOS Biology.

Your manuscript has now been evaluated by the PLOS Biology editorial staff as well as by an academic editor with relevant expertise and I am writing to let you know that we would like to send your submission out for external peer review.

**Important**: Please also see below for further information regarding completing the MDAR reporting checklist. The checklist can be accessed here: https://plos.io/MDARChecklist

Please re-submit your manuscript and the checklist, within two working days, i.e. by Jul 21 2019 11:59PM.

Kind regards,

Di Jiang

PLOS Biology

INFORMATION REGARDING THE REPORTING CHECKLIST:

PLOS Biology is pleased to support the "minimum reporting standards in the life sciences" initiative (https://osf.io/preprints/metaarxiv/9sm4x/). This effort brings together a number of leading journals and reproducibility experts to develop minimum expectations for reporting information about Materials (including data and code), Design, Analysis and Reporting (MDAR) in published papers. We believe broad alignment on these standards will be to the benefit of authors, reviewers, journals and the wider research community and will help drive better practise in publishing reproducible research. 

We are therefore participating in a community pilot involving a small number of life science journals to test the MDAR checklist. The checklist is intended to help authors, reviewers and editors adopt and implement the minimum reporting framework. 

IMPORTANT: We have chosen your manuscript to participate in this trial. The relevant documents can be located here:

MDAR reporting checklist (to be filled in by you): https://plos.io/MDARChecklist

**We strongly encourage you to complete the MDAR reporting checklist and return it to us with your full submission, as described above. We would also be very grateful if you could complete this author survey:

https://forms.gle/seEgCrDtM6GLKFGQA

Additional background information:

Interpreting the MDAR Framework: https://plos.io/MDARFramework

Please note that your completed checklist and survey will be shared with the minimum reporting standards working group. However, the working group will not be provided with access to the manuscript or any other confidential information including author identities, manuscript titles or abstracts. Feedback from this process will be used to consider next steps, which might include revisions to the content of the checklist. Data and materials from this initial trial will be publicly shared in September 2019. Data will only be provided in aggregate form and will not be parsed by individual article or by journal, so as to respect the confidentiality of responses. 

Please treat the checklist and elaboration as confidential as public release is planned for September 2019.

We would be grateful for any feedback you may have.

---

## [Decision Letter · Decision Letter 1]

19 Aug 2019

Dear Dr Nightingale,

Thank you very much for submitting your manuscript "Dynamic trafficking and turnover of Jam-C is essential for endothelial cell migration" for consideration as a Research Article at PLOS Biology. Your manuscript has been evaluated by the PLOS Biology editors, an Academic Editor with relevant expertise, and by three independent reviewers.

In light of the reviews (below), we will not be able to accept the current version of the manuscript, but we would welcome resubmission of a revised version that addresses the reviewers' comments. Our Academic Editor indicates that the comments of the reviewers are of high quality and are very reasonable because they mostly request better description of the tools and data already presented in the manuscript. S/he advises that you address experimentally comment 1 of reviewer 1 on characterizing better your fusion construct, and all the comments of the reviewers that request clarifications, quantifications, explanations of control experiments or better discussion/wording. We cannot make any decision about publication until we have seen the revised manuscript and your response to the reviewers' comments. Your revised manuscript is also likely to be sent for further evaluation by the reviewers.

Your revisions should address the specific points made by each reviewer. Please submit a file detailing your responses to the editorial requests and a point-by-point response to all of the reviewers' comments that indicates the changes you have made to the manuscript. In addition to a clean copy of the manuscript, please upload a 'track-changes' version of your manuscript that specifies the edits made. This should be uploaded as a "Related" file type. You should also cite any additional relevant literature that has been published since the original submission and mention any additional citations in your response. 

Before you revise your manuscript, please review the following PLOS policy and formatting requirements checklist PDF: http://journals.plos.org/plosbiology/s/file?id=9411/plos-biology-formatting-checklist.pdf. It is helpful if you format your revision according to our requirements - should your paper subsequently be accepted, this will save time at the acceptance stage.

Please note that as a condition of publication PLOS' data policy (http://journals.plos.org/plosbiology/s/data-availability) requires that you make available all data used to draw the conclusions arrived at in your manuscript. If you have not already done so, you must include any data used in your manuscript either in appropriate repositories, within the body of the manuscript, or as supporting information (N.B. this includes any numerical values that were used to generate graphs, histograms etc.). For an example see here: http://www.plosbiology.org/article/info%3Adoi%2F10.1371%2Fjournal.pbio.1001908#s5.

For manuscripts submitted on or after 1st July 2019, we require the original, uncropped and minimally adjusted images supporting all blot and gel results reported in an article's figures or Supporting Information files. We will require these files before a manuscript can be accepted so please prepare them now, if you have not already uploaded them. Please carefully read our guidelines for how to prepare and upload this data: https://journals.plos.org/plosbiology/s/figures#loc-blot-and-gel-reporting-requirements.

Upon resubmission, the editors will assess your revision and if the editors and Academic Editor feel that the revised manuscript remains appropriate for the journal, we will send the manuscript for re-review. We aim to consult the same Academic Editor and reviewers for revised manuscripts but may consult others if needed.

We expect to receive your revised manuscript within two months. Please email us (plosbiology@plos.org) to discuss this if you have any questions or concerns, or would like to request an extension. At this stage, your manuscript remains formally under active consideration at our journal; please notify us by email if you do not wish to submit a revision and instead wish to pursue publication elsewhere, so that we may end consideration of the manuscript at PLOS Biology.

When you are ready to submit a revised version of your manuscript, please go to https://www.editorialmanager.com/pbiology/ and log in as an Author. Click the link labelled 'Submissions Needing Revision' where you will find your submission record. 

Sincerely,

Di Jiang, PhD

Associate Editor

PLOS Biology

Reviewer remarks:

Reviewer #1: Kostelnik, et al. present interesting data that JAM-C is internalized in a ubiquitin-dependent manner along with molecules regulating permeability such as VE-cadherin and NRP1 and NRP2, but not with molecules regulating leukocyte transmigration such as PECAM-1, CD99, and JAM-A, and under conditions in which endothelial cells move, this internalization is critical for movement by reducing the amount of JAM-C in the lateral junctions to allow adjacent endothelial cells (EC) to break their bonds and slide past each other. They use a clever technique to label membrane molecules neighboring JAM-C and distinguish those on the surface from those internalized. The data are for the most part convincing. However, I have some reservations regarding the central interpretation that should be reconciled prior to publication.

MAJOR COMMENTS

The major issue is that the authors assume that the pathway taken by JAM-C-HRPout is the same as taken for native JAM-C. Their justification for this is, “Tagging of Jam-C at this point has previously been shown to have no effect on ligand binding or localisation.” They refer to a previous paper (Mol Biol Cell 16, 4992-5003, 2005) in which they show that placing eGFP—a molecule of approximately the same size as HRP--in that position still allows JAM-C to localize to the junctions appropriately when transfected into MDCK cells or CHO fibroblasts. However, they never studied turnover of the construct. JAM-C has an apparent Mr of 29 - 45 kD depending on the report. HRP has an Mr of 44 kD. It is quite possible that placing a molecule the size of JAM-C or larger into the JAM-C extracellular domain would affect its turnover and its ability to recycle. This raises the question of whether they are studying the natural trafficking pathway of JAM-C or the degradation pathway of a distorted molecule. The half-life of JAM-C-GFPout seems to be about 12 hours in Fig. 4D, which is rather short for a membrane glycoprotein, particularly a member of the Ig superfamily, which in a recent study was found to have a median half-life of 20 hours (Chem. Sci., 2017, 8, 268). To be certain that their constructs are following the normal turnover pathways, they need to compare the turnover of JAM-C-HRPout with that of native JAM-C in the same experiment. They could do this using the same procedure as in Fig. 4, or more accurately using metabolically labeled then “chased” molecules followed by immunoprecipitation and western blot. If the turnover of JAM-C-HRPout is considerably faster than that of native JAM-C, the observations that JAM-C is internalized with a subset of surface interacting proteins and that enhanced degradation of JAM-C facilitates cell migration are still interesting observations worthy of publication, but the interpretation of the data will need to be changed.

 Fig. 2 The antibody against JAM-C used in A-C is a polyclonal antibody, which can cross-link JAM-C and increase its rate of internalization. What happens if a monoclonal antibody is used?

Fig. 3E. What is “mock” conditions? I can’t find this explained anywhere. Why is so much material pulled down under “mock” conditions in E but not seen in the corresponding proteomic experiments in 3F?

Figs. 5, 6, 7 Considering that ubiquitylation of membrane proteins is a well-known and accepted mechanism for targeting them for degradation, the data of these figures does not seem very surprising or significant compared to the proteomic data of Fig 3. The E3 ligase for JAM-C was not previously known, but it is not surprising that one exists. The authors could consider shortening this part of the manuscript or relegating it to Supplementary Materials and focus on the significance of their primary observation and the experiment of Figure 8.

Fig. 8 shows that inhibition of ubiquitination and turnover of JAM-C blocks migration. The authors propose that this is due to more JAM-C at the junctions. Previous figures show that blocking ubiquitylation of JAM-C increases JAM-C-GFPout levels and decreases the number of fluorescent vesicles, but the intensity of JAM-C at the junctions does not appear to be different. Can the authors quantify junctional JAM-C? 

Fig. 9 The video provided does not make a good case that internalization of JAM-C-GFPout is at the sides of the cells as opposed to at the front of the migrating cell. 

MINOR COMMENTS

Almost none of the western blots have molecular size labels. They should.

Fig. 1 Were the cells permeabilized to be able to see punctae?

Line 297 TNFa is a known permeability agent; IL-1b is not. Therefore, instead of being stimulus specific, a more circumspect interpretation would be that this change is seen with treatments that reduce barrier function.

It is a bit disingenuous in the schematic diagrams in multiple figures to represent the HRP and GFP as small circles compared to the large JAM-C sausage, when in fact HRP and GFP are about the same size or larger than the JAM-C extracellular domains. They should be drawn to scale.

I am curious why the manuscript refers to the molecule as “Jam-C” when the rest of the literature as well as Dr. Aurrand-Lions, one of the pioneers of this molecule, calls it “JAM-C”. 

Summary: This is an interesting manuscript. The authors should perform the experiments suggested above, which should not be difficult in order to better understand the phenomena they are studying. The statistics appear to be adequate and supplemental data are appropriate. The manuscript is generally well-written, but quite a few experimental details are missing from the Methods and figure legends. This should be reviewed carefully when revising.

Reviewer #2: This manuscript reports about recycling mechanisms of the adhesion molecule JAM-C in endothelial cells. It is shown that JAM-C is constitutively endocytosed and recruited to multi-vesicular bodies leading to its degradation. Internalization is boosted by junctional disassembly and the cytokine TNF-�. A method of proximity biotinylation based on integrating HRP into the JAM-C protein was used to identify candidate proteins by proteomics, which might be within a 20 nm distance to JAM-C on the cell surface and in endocytic vesicles. A group of such candidates were found within the vicinity of JAM-C in both locations, possibly trafficking parallel with JAM-C, others were only in the vicinity of JAM-C at the cell surface. A lysine residue was reported to be involved in JAM-C turn over. Mutating all four lysine residues in the cytoplasmic domain blocked its ubiquitylation as well as its targeting to multi-vesicular bodies. Another approach to biotinylate proteins within the cytoplasmic tail of JAM-C was based on inserting APEX-2 into the cytoplasmic tail. Combining this with the mutation of the four lysine residues allowed to identify the E3 ligase Cbl as candidate of a protein adjacent to JAM-C during trafficking. Silencing of Cbl inhibited ubiquitylation of JAM-C and increased its level in endothelial cells, however, did not affect its levels at cell contacts. Finally, mutating the four lysine residues in JAM-C impaired recovery in scratch wound assays of endothelial monolayers.

The study is interesting and establishes a role for JAM-C ubiquitylation for its targeting to the degradation pathway which seems to be relevant for mechanisms by which JAM-C influences endothelial cell contacts and or endothelial cell migration. The results presented in this paper are likely to contribute to the understanding of the role of JAM-C in angiogenesis and possibly other functions of JAM-C. The paper falls short of explaining how impairment of targeting JAM-C to the degradation pathway might influence cell migration. While this may be asked too much, some other aspects need to be improved.

1) The HRP based proximity labeling assay was used in combination with streptavidin/biotin based pull down experiments and proteomics analysis to identify a series of proteins which are candidates for close neighbors within the vicinity of JAM-C. Verification of these candidates was attempted, but the results in supplementary figure 2 are not convincing. The proteomics results suggest that biotinylation of VE-cadherin, NRP-1 and NRP-2, triggered by JAM-C-HRP, is not completely killed by ascorbate, thus it occurs at the cell surface and in intracellular vesicles. This is indeed verified in suppl. figure 2B. However, the proteomics data suggest, that biotinylation of PECAM-1 and CD99 was not seen at all and for JAM-A there was a weak signal without ascorbate and no signal with ascorbate. The note “bands not visible due to low antibody sensitivity” as stated within the figure is not convincing. As it stands right now, no statement can be made about PECAM-1 and CD99 (they can even not be considered as confirmed candidates for proteins in the vicinity of JAM-C) and for JAM-A the overall signal (outside plus inside the cell) is so weak that a comparison between outside and inside (with and without ascorbate) cannot be made. The results shown in suppl. figure 2A have to be done in a more sensitive way to obtain convincing and meaningful results.

2) For several experiments “mock” conditions were shown, but it was not stated what these negative control conditions were. For example, Fig. 3E and 3F, does mock mean a mock transfection with a JAM-C without HRP? Same for Fig. 6C and 6E. What about suppl fig. 2A and B, does “mock” mean no hydrogenperoxide, or no JAM-C-HRP or no biotinylation reagent?

3) Silencing of JAM-C did not lead to enhanced expression levels of JAM-C at cell contacts (Fig. 7F). Why then do the authors discuss that “the change in receptor recycling could prematurely increase the strength of homo and heterotypic junctional interactions with neighboring cells preventing cell movement” (bottom of page 17)? This argument should probably be omitted. Furthermore, the cartoon in figure 9B also seems to suggest that inhibition of JAM-C ubiquitylation would lead to more JAM-C at cell contacts, but this is not observed in reality (Fig. 7F). Therefore, the cartoon should be omitted or redrawn in a way, that is not in conflict with the results of figure 7F. Furthermore, the discussion should mention that lack of JAM-C ubiquitylation might affect cell migration by affecting integrins. In the light of their results showing co-trafficking of integrins with JAM-C and in the light of publications analyzing possible interactions between JAM-C and integrins (such as reference 13 and 48), these are obvious possibilities.

Reviewer #3: With pleasure I have read the manuscript of Kostelnik et al describing a role for Jam-C protein trafficking in endothelial cell migration. The authors have performed imaging based analysis of Jam-C dynamics as well as elegant proteomic approaches to discern Jam-C function at the cell’s surface or once internalized. This has lead to the finding that Jam-C is ubiquitinated by CBL and that proper degradation of internalized Jam-C is needed for endothelial migration in scratch wound assays. In addition, the trafficking of Jam-C seems to occur in distinct vesicles compared to some of the other well-known endothelial receptors of the lateral recycling compartment. Which reveals an alternative mode of junction regulation Jam-C (and co-trafficking VE-cadherin). Overall the experiments have been well controlled and represent exciting new findings that are of importance for the field and would fit well within the scope of Plos Biology. I have a few concerns, which I encourage the authors to address. 

• The authors conclude that Jam-C recycling is important, however there is no direct evidence to suggest that this is the case. Instead all experiments have been focused on internalization, ubiquitination and proteosomal degradation. Therefore statements along those lines should be avoided. Or the authors have to try to proof that Jam-C is recycled toward the plasma membrane and that this is the mechanism how Jam-C promote scratch wound migration. 

• Figure 7: Is internalization of endogenous Jam-C affected by CBL? Pulse chase experiments with the anti-Jam-C antibody should be compared in control and siRNA treated HUVECs. Is the quad-K mutant of Jam-C able to rescue migration defects in CBL knockdown cells?

• Please confirm that the Quad lysine mutated Jam-C variant is not ubiquitinated anymore by CBL.

• Figure 9 and text page 15: “We noted large Jam-C positive vesicles forming at junctional sites on either side of the cell’s leading edge”. Although the data in this figure indeed indicate that Jam-C is internalized in leader cells in the scratch assay, there is no comparison made with follower cells or cells in non-migrating endothelial cells to be able to determine whether Jam-C internalization is specific for, or enriched at the leading edge. 

• Does mutating the lysines in Jam-C tail affect only intracellular trafficking or also overall surface levels of Jam-C? Please analyse the amount of Jam-C surface levels of the wild type protein versus the Quad-K mutant. 

• The authors claim to have shown a new trafficking machinery in control of Jam-C. Although the mass spectrometry results indeed have identified many proteins to be in the vicinity of Jam-C, the study continues only with knockdown studies with CBL. It is more appropriate to mention that CBL-mediated ubiquitination controls Jam-C. Unless the authors can show that other identified trafficking regulators are functionally involved in controlling trafficking of Jam-C. 

Minor Comments

• Figure 2A, B: is the antibody used to determine kinetics of Jam-C internalization not interfering with the function of Jam-C? Analysis of VE-cadherin stainings and scratch wound migration the presence of the antibody should be provided to confirm that the analysis based on this antibody closely resembles trafficking of the endogenous Jam-C protein.

• Figure 2E: Is the observed increased number of Jam-C positive vesicles upon TNF treatment due to decreased proteosomal degradation or due to increased internalization of Jam-C from the plasma membrane?

• Figure 2G: The authors conclude that TNFa enhances internalization of Jam-C in Fig 2D. What happens to the dynamics of Jam-C-GFP after treatment with TNFa, are more membraneous structures observed under those conditions and does that explain the increased number of Jam-C vesicles? 

• In the text there is a big emphasis on the role of Jam-C in leukocyte transmigration. However, the data presented are regarding the dynamics of Jam-C in normal endothelial monolayers, scratch assays or TNF-treated endothelium. I recommend to either prove that CBL-mediated Jam-C protein trafficking is involved in transmigration of leukocytes across endothelium or to change the focus of the manuscript to better reflect the current results. 

• The presentation of immunofluorescence images would benefit from higher intensities (i.e. Figure 1B,C;2C; 7F.

• I encourage the authors to improve the presentation of the figures by marking performed immuno-stainings in the figure itself rather than in the figure legends only. 

• Figure 2B; how was the junctional area defined to be able to quantify whether Jam-C intensity levels?

• What happens to VE-cadherin trafficking, which is also internalized through a similar mode of tubular endocytosis from remodeling junctions, once the lysines in Jam-C are mutated and preventing its proteosomal degradation. Does the trafficking of other receptors remain unaffected or is their a co-dependent mode of internalization and endocytosis?

---

## [Editor Report · Decision Letter 2]

29 Oct 2019

Dear Dr Nightingale,

Thank you for submitting your revised Research Article entitled "Dynamic trafficking and turnover of JAM-C is essential for endothelial cell migration" for publication in PLOS Biology. I have now obtained advice from the academic editor who has assessed your revision.

Based on our academic editor's evaluation, we're delighted to let you know that we're now editorially satisfied with your manuscript. However before we can formally accept your paper and consider it "in press", we also need to ensure that your article conforms to our guidelines; two of which are described below under DATA POLICY and are marked with "***IMPORTANCE: ". A member of our team will be in touch shortly with a set of requests. As we can't proceed until these requirements are met, your swift response will help prevent delays to publication.

Please note that you have the opportunity to make the peer review history publicly available. The record will include editor decision letters (with reviews) and your responses to reviewer comments. Our academic editor felt that your rebuttal was very nice and useful to readers and that the paper would benefit from having it published along with the reviews. We would like to encourage you to opt in.

Sincerely,

Di Jiang, PhD

Associate Editor

PLOS Biology

ETHICS STATEMENT:

You may remove the current Ethics Statement (copied below) from the submission form.

"No human participants, specimens or tissue samples, or vertebrate animals, embryos or tissues were used as part of this study."

DATA POLICY:

***IMPORTANCE: Regardless of the method selected, please ensure that you provide the individual numerical values that underlie the summary data displayed in the following figure panels: 1D, 2BCE, 3F, 4D, 5DE, 6E, 7CDEG, 8BC, S1BDF, S2AB, S3C, as they are essential for readers to assess your analysis and to reproduce it. ***IMPORTANCE: Please also ensure that figure legends in your manuscript include information on where the underlying data can be found. You may say in the relevant figure legends: e.g., "Underlying data are found in S1 Data.".

For manuscripts submitted on or after 1st July 2019, we require the original, uncropped and minimally adjusted images supporting all blot and gel results reported in an article's figures or Supporting Information files. We will require these files before a manuscript can be accepted so please prepare them now, if you have not already uploaded them. Please carefully read our guidelines for how to prepare and upload this data: https://journals.plos.org/plosbiology/s/figures#loc-blot-and-gel-reporting-requirements.

In compliance with data protection regulations, you may request that we remove your personal registration details at any time. (Use the following URL: https://www.editorialmanager.com/pbiology/login.asp?a=r). Please contact the publication office if you have any questions.

---

## [Editor Report · Decision Letter 3]

14 Nov 2019

Dear Dr Nightingale,

On behalf of my colleagues and the Academic Editor, Anna Akhmanova, I am pleased to inform you that we will be delighted to publish your Research Article in PLOS Biology. 

Early Version

PRESS 

Kind regards,

Hannah Harwood

Publication Assistant, 

PLOS Biology

on behalf of

Di Jiang,

Associate Editor

PLOS Biology